

# Comparative anatomy of the middle ear in some lizard species with comments on the evolutionary changes within Squamata

Paola María Sánchez-Martínez[1], Juan D. Daza[2] and Julio Mario Hoyos[1,3]

[1] Laboratorio de Sistemática Morfológica y Biogeografía de Vertebrados, Departamento de Biología Facultad de Ciencias, Pontificia Universidad Javeriana, Bogotá, Cundinamarca, Colombia
[2] Department of Biological Sciences, Sam Houston State University, Huntsville, Texas, United States
[3] Unidad de Ecología y Sistemática (UNESIS), Departamento de Biología, Facultad de Ciencias, Pontificia Universidad Javeriana, Bogotá, Cundinamarca, Colombia

Corresponding authors
Paola María Sánchez-Martínez, paola.sanmart@gmail.com
Julio Mario Hoyos, jmhoyos@javeriana.edu.co

## ABSTRACT

The skeleton of the middle ear of lizards is composed of three anatomical elements: columella, extracolumella, and tympanic membrane, with some exceptions that show modifications of this anatomy. The main function of the middle ear is transforming sound waves into vibrations and transmitting these to the inner ear. Most middle ear studies mainly focus on its functional aspects, while few describe the anatomy in detail. In lizards, the morphology of the columella is highly conservative, while the extracolumella shows variation in its presence/absence, size, and the number of processes present on the structure. In this work, we used diaphanized and double-stained specimens of 38 species of lizards belonging to 24 genera to study the middle ear's morphology in a comparative framework. Results presented here indicate more variation in the morphology of the extracolumella than previously known. This variation in the extracolumella is found mainly in the pars superior and anterior processes, while the pars inferior and the posterior process are more constant in morphology. We also provide new information about the shape of gekkotan extracolumella, including traits that are diagnostic for the iguanid and gekkonid middle ear types. The data collected in this study were combined with information from published descriptive works. The new data included here refers to the length of the columella relative to the extracolumella central axis length, the general structure of the extracolumella, and the presence of the internal process. These characters were included in ancestral reconstruction analysis using Bayesian and parsimony approaches. The results indicate high levels of homoplasy in the variation of the columella-extracolumella ratio, providing a better understanding of the ratio variation among lizards. Additionally, the presence of four processes in the extracolumella is the ancestral state for Gekkota, Pleurodonta, and Xantusiidae, and the absence of the internal processes is the ancestral state for Gekkota, Gymnophthalmidae, and Scincidae; despite the fact that these groups convergently develop these character states, they could be used in combination with other characters to diagnose these clades. The posterior extension in the pars superior and an anterior process with some small and sharp projections is also a diagnostic trait for Gekkota. A more accurate description of each process of the extracolumella and

its variation needs to be evaluated in a comprehensive analysis, including a greater number of species. Although the number of taxa sampled in this study is small considering the vast diversity of lizards, the results provide an overall idea of the amount of variation of the middle ear while helping to infer the evolutionary history of the lizard middle ear.

## INTRODUCTION

The ear is a complex system that performs a dual function—equilibrium and hearing. In reptiles, the ear has been described in three divisions: the outer, middle, and inner ear (*Baird, 1970*). The outer ear includes the meatal cavity, closure muscles, and modifications of skin that detect sound waves and conduct them to the middle ear. In the middle ear of lizards (in most species of lizards composed of the tympanic membrane, extracolumella, and columella) the sound waves are transformed into vibrations, which are transmitted to the inner ear. The inner ear also is formed by the membranous or endolymphatic labyrinth where the sensorial organs are located, and the perilymphatic labyrinth is an area of fluid-filled cavities in which the movements continue as fluid oscillations, impacting the cochlea (*Baird, 1960*, *1970*; *Wever, 1978*). Most of the studies around the lizard ear are focused on the study of processes of conductivity of sound, and the electrophysiological aspects of the inner ear (e.g., *Shute & Bellairs, 1953*; *Baird, 1960*; *Wever et al., 1963*; *Schmidt, 1964*; *Wever et al., 1965*; *Baird, 1967*; *Suga & Campbell, 1967*; *Wever, 1967*, *1970*; *Baird & Marovitz, 1971*; *Wever, 1971*; *Manley, 1972a*; *Wever & Gans, 1972*; *Miller, 1974*; *Werner, 1976*; *Manley, 2000*; *Werner & Igić, 2002*; *Wibowo, Brockhausen & Köppl, 2009*; *Manley, 2011*). The standard approach of studies on the middle ear has been mainly focused on investigating the functional aspects of the transformation of sound waves into vibrations, with some work describing a few morphological features (e.g., *Wever & Peterson, 1963*; *Wever & Werner, 1970*; *Manley, 1972b*; *Werner & Wever, 1972*; *Wever, 1973*; *Manley, 2011*; *Han & Young, 2016*). Other studies, although less common, have concentrated specifically on the anatomy of the middle and outer ear (e.g., *Versluys, 1898*; *Earle, 1961a*; *Earle, 1961b*; *Earle, 1961c*; *Posner & Chiasson, 1966*; *Iordansky, 1968*; *Wever, 1978*). The studies that could be considered the most relevant contributions to knowledge of the middle ear in lizards are those by *Versluys (1898)* and *Wever (1978)*. *Versluys (1898)* shared essential information about the morphology of the structures and associated muscles. *Wever (1978)* contributed to the knowledge of the function of the inner ear, describing details of the structures of the middle and outer ear and its taxonomic distribution, information that has been used in cladistics studies (e.g., *Kluge, 1987*).

In lizards, the most common pattern of the middle ear (Fig. 1) is a simple structure composed of the columella and extracolumella that are suspended in the tympanic cavity, and the tympanic membrane. Some groups also show the internal process, which is an

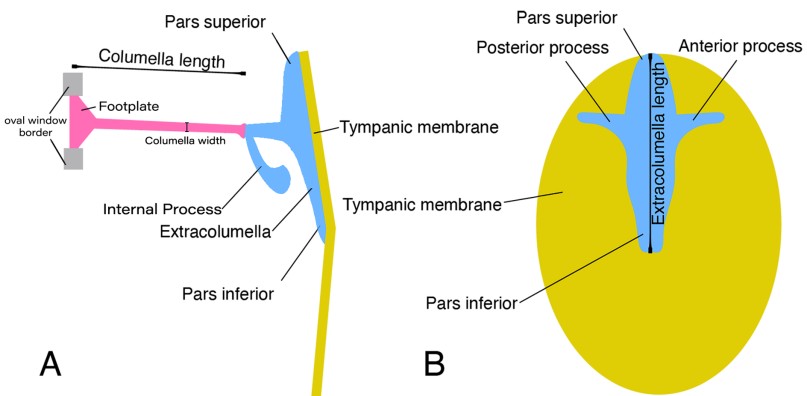

**Figure 1 Schematic representation of the middle ear of lizards.** Illustrative sketch of the structures that conform the middle ear of lizards. (A) Middle ear (from the posterior view of the skull); (B) extracolumella and tympanic membrane (from the lateral view of the skull). Modified from *Mason & Farr (2013)*.                                                           

additional middle ear cartilaginous element associated with the extracolumella (*Versluys, 1898*; *Baird, 1970*; *Wever, 1978*; *Saunders et al., 2000*). The columella (Fig. 1A) is a slender rod whose main part is osseous, and its distal end is cartilaginous. The proximal end is formed by a footplate; this end of the bone inserts itself into the oval window which is the opening of the otic capsule leading to the inner ear and connects with the cochlea. At the distal end of the columella, the bone is connected to the extracolumella. The extracolumella (Figs. 1A and 1B) is a cartilaginous structure forming a main shaft that shows a variable number of processes (two to four), namely: pars superior and pars inferior, the anterior and posterior processes. These processes meet the internal surface of the tympanic membrane in a cruciform arrangement. The principal extracolumellar processes are the pars superior and pars inferior, which form a vertical shaft whose function is to transmit the vibrations and stretch and tense the tympanic membrane. In most of the species, the pars superior and inferior are associated with the extracolumellar and intratympanic ligaments, respectively. Also, in most of the gekkotans, the pars superior is associated with the extracolumellar muscle that probably exercises tension on the membrane and the other structures of the middle ear (*Wever & Werner, 1970*; *Wever, 1978*). The anterior and posterior processes arise from the pars superior and pars inferior and are smaller than the structures from which they originate, sometimes being poorly defined or absent in some species (*Wever, 1978*). When these extracolumellar processes are developed, they attach the extracolumella to the tympanic membrane to reduce movements of the extracolumella and help to tense the membrane surface (*Baird, 1970*; *Wever, 1978*; *Saunders et al., 2000*). The internal process is a complementary extracolumellar structure that is present only in iguanians and related species. This process originates from the extracolumella and serve to link the extracolumella to the quadrate bone. In species where the internal process is absent, the support of the columellar system is given by a fold of mucous membrane (*Wever, 1978*). The extracolumella is the element of the middle ear in lizards that displays the most morphological variation. This variation tends to occur in the shape and number of the extracolumellar processes, the presence or
absence of the internal process, and the type of the connection between the columella and the extracolumella (*Wever, 1978*).

Based on the overall morphology, *Wever & Werner (1970)* defined three main patterns of middle ears in lizards, namely the gekkonid, iguanid, and scincid types. Additionally, different forms that do not correspond to the previous patterns were considered as "divergent" types, which mostly were morphologies that departed the iguanid type (*Wever, 1978*). These three standard types exhibit the same primary structure described above but differ in some details associated with both presence and form of certain structures. In the iguanid type (*Wever, 1978*, Fig. 6–10), the most generalized type in lizards, there is an additional cartilaginous shaft termed the "internal process" by *Versluys (1898)*, which arises from the extracolumellar shaft and expands dorsally and anteriorly to attach to the quadrate bone. In the gekkonid type (*Wever, 1978*, Fig. 6–30), there is no internal process, but there is a tympanic muscle called the "extracolumellar muscle" (*Wever & Werner, 1970*), that runs from the distal edge of the pars superior to the ceratohyal process. The scincid type (*Wever, 1978*, Fig. 6–42) lacks both the internal process and the tympanic muscle; and the divergent types show features that do not match with any of the aforementioned types (*Wever, 1978*).

The middle ear has evolved independently several times in vertebrates (*Lombard & Bolt, 1979*; *Clack, 1997*; *Clack, 2002*; *Manley, 2010*). In the stem reptiles, the tympanum is absent, and the stapes is bulky. The evolutive changes in the stapes resulted in unique middle-ear morphologies present in each order of living reptiles (*Saunders et al., 2000*). In lizards, the studies presented by *Versluys (1898)*, *Olson (1966)*, and *Baird (1970)* made anatomical comparisons of the outer and middle ear among taxa making some evolutionary assumptions. According to *Olson (1966)*, the middle ear is associated with the masticatory apparatus and is therefore highly susceptible to adaptive modifications and, although some morphological types are conservative, others are rather diverse. Thus, the middle ear structures could prove to be useful in providing phylogenetic information within major morphological types, but not when relationships between these types are considered (*Olson, 1966*). *Baird (1970)* suggests that in most terrestrial and arboreal lizards, the middle ear corresponds to the iguanid pattern, but it is common to find related taxa that show morphological variations correlated to other features of the ear, or variations that may relate more directly to habits or habitats. However, this kind of affirmation is preliminary because the diversity of morphologies of the external and middle ear across lizards is barely understood and requires further investigation (*Wever, 1968*; *Baird, 1970*). The main objective of this study is to describe the morphological variation of the middle ear in "lizards", using samples from four of the main taxonomic groups (Gekkota, Iguania, Lacertoidea, and Scincoidea (*Zheng & Wiens, 2016*)), and character trait mapping methods to propose a preliminary scenario of middle ear evolution.

Since the word "lizard" refers to a paraphyletic group relative to snakes, we must clarify that by using this term we refer to squamates that are not snakes (i.e., Iguania, Gekkota, Scincoidea, Lacertoidea [including Amphisbaena], and Anguimorpha (excluding snakes)). Despite the paraphyletic status of "lizards", it makes sense for us to study them

as a whole considering their shared similarities in middle ear structures and their differences with snakes.

## MATERIALS & METHODS

### Comparative anatomy

We examined the middle ear of cleared and double-stained specimens of 38 species of lizards, belonging to 24 genera and 12 families (Table 1). We recognize that this number of species examined is a small percentage of the totality of species of lizards described, however this small sample size is adequate to produce an initial assessment of the morphological differences in the middle ear of lizards. The specimens examined belong to the Colección Herpetológica del Museo Javeriano de Historia Natural Lorenzo Uribe, S.J.—MUJ at Pontificia Universidad Javeriana (Bogotá, Colombia), Colección Herpetológica del Instituto de Ciencias Naturales—ICN at Universidad Nacional de Colombia (Bogotá, Colombia), Museo de Herpetología de la Universidad de Antioquia— MHUA (Medellín, Colombia), and the Museu de Zoologia da Universidade de São Paulo— MZUSP (São Paulo, Brazil). Voucher specimen information is provided in Table S1. The middle ears of the species studied were described following the nomenclature proposed by *Wever (1978)* and analyzed in a comparative framework with the data available in the literature. The summary of the variation described is presented in Tables 2 and 3.

As a note on taxonomy within this paper, we have considered the genus *Mabuya* in the broad sense. The genus *Mabuya* was extensively rearranged in 2012, and here we examined species from the clade referred to as "American Mabuyas," which now encompasses eight genera (*Hedges & Conn, 2012*). In this study, we used specimens from two of these American genera (*Copeoglossum nigropunctatum* and *Marisora falconensis*) together with other undescribed species, but for simplicity, we have referred all of them to the genus *Mabuya* sp.

### Ancestral reconstruction

Character states were coded from direct observations of the material described and from published data. The sources of the information published for each species included in the analysis are given in Table 4. In order to reconstruct the evolutionary changes, the morphological characters defined were optimized on the phylogenetic hypothesis based on molecular data proposed by *Zheng & Wiens (2016)*, using maximum parsimony (MP) and Bayesian approaches. The parsimony analysis used equal weighting, the characters were considered as unordered and the analysis was performed using MESQUITE 3.5 (*Maddison & Maddison, 2018*). The Bayesian analysis used the "ARD" (backward & forward rates between states) and "ER" (single-rate) models, and was conducted using R 4.0.2 (*R Core Team, 2020*) and the phytools package (*Revell, 2012*). To perform the parsimony analysis, we pruned the tree to include only the species studied here, and in some cases, we edited terminal names following two rules: (1) if several species from a single genus had the same character state, these were collapsed into a single terminal with the genus name (the list of species collapsed and their corresponding terminal taxon are

**Table 1  Species and number of specimens examined.**

| Group | Family | Genus | Species | Number of Specimens |
|---|---|---|---|---|
| Gekkota | Gekkonidae | *Hemidactylus* | *H. brasilianus* | 1 |
| | | *Phelsuma* | *P. madagascariensis* | 1 |
| | Phyllodactylidae | *Tarentola* | *T. mauritanica* | 1 |
| | | *Thecadactylus* | *T. rapicauda* | 1 |
| | Pygopodidae | *Lialis* | *L. jicari* | 1 |
| | Sphaerodactylidae | *Gonatodes* | *G. albogularis* | 1 |
| | | | *G. concinnatus* | 1 |
| Iguania | Agamidae | *Acanthocercus* | *A. atricollis* | 1 |
| | | *Leiolepis* | *L. belliana* | 1 |
| | | *Stellagama* | *S. stellio* | 1 |
| | Dactyloidae | *Anolis* | *A. antonii* | 2 |
| | | | *A. auratus* | 2 |
| | | | *A. chrysolepis* | 2 |
| | | | *A. fuscoauratus* | 1 |
| | | | *A. maculiventris* | 4 |
| | | | *A. mariarum* | 3 |
| | | | *A. tolimensis* | 2 |
| | | | *A. trachyderma* | 2 |
| | | | *A. ventrimaculatus* | 3 |
| | Hoplocercidae | *Hoplocercus* | *H. spinosus* | 1 |
| | | *Morunasaurus* | *M. groi* | 1 |
| | Tropiduridae | *Stenocercus* | *S. erythrogaster* | 1 |
| | | *Tropidurus* | *S. trachycephalus* | 2 |
| | | | *T. pinima* | 1 |
| Lacertoidea | Gymnophthalmidae | *Anadia* | *A. bogotensis* | 4 |
| | | *Gelanesaurus* | *G. cochranae* | 1 |
| | | *Loxopholis* | *L. rugiceps* | 1 |
| | | *Neusticurus* | *N. medemi* | 1 |
| | | *Pholidobolus* | *P. montium* | 2 |
| | | | *P. vertebralis* | 1 |
| | | *Riama* | *R. striata* | 3 |
| | | *Tretioscincus* | *T. bifasciatus* | 1 |
| | Teiidae | *Cnemidophorus* | *C. lemniscatus* | 1 |
| | Lacertidae | *Acanthodactylus* | *A. cf. schmidti* | 1 |
| Scincoidea | Scincidae | *Mabuya* | *M. falconensis* | 1 |
| | | | *M. nigropunctatum* | 2 |
| | | | *Mabuya* sp. 1 | 2 |
| | | | *Mabuya* sp. 2 | 3 |

**Note:**
The taxonomic classification follows *Zheng & Wiens (2016)*.

provided in Table S2); (2) if one or more examined taxa were not included in the molecular phylogenetic analysis, these taxa were included as terminals in a polytomy, assuming that the genera are monophyletic. Features with unknown character states were treated as missing "?", and inapplicable characters as dash "−". To conduct the Bayesian analysis, we pruned the topology by collapsing the genera without data to a single terminal for family.

**Table 2 Characterization of the morphological variation of the columella, and the joint with the extracolumella.**

| Species | Columella | | | Joint of stapes |
|---|---|---|---|---|
| | Stapedial foramen | *Length of the columella | Widening of the osseous distal end | Connective tissue |
| GEKKOTA | | | | |
| Gekkonidae | | | | |
| *Hemidactylus brasilianus* | present | equal | absent | absent |
| *Phelsuma madagascariensis* | present | equal | present | absent |
| Phyllodactylidae | | | | |
| *Tarentola mauritanica* | present | longer | absent | surrounding the joint |
| *Thecadactylus rapicauda* | absent | equal | absent | absent |
| Pygopodidae | | | | |
| *Lialis jicari* | absent | shorter | present | absent |
| Sphaerodactylidae | | | | |
| *Gonatodes albogularis* | present | shorter | absent | absent |
| *Gonatodes concinnatus* | present | shorter | absent | absent |
| IGUANIA | | | | |
| Agamidae | | | | |
| *Acanthocercus atricollis* | ? | longer | present | surrounding the joint |
| *Leiolepis belliana* | ? | ? | absent | absent |
| *Stellagama stellio* | ? | ? | ? | ? |
| Dactyloidae | | | | |
| *Anolis antonii* | absent | equal | present | between the joint |
| *Anolis auratus* | absent | equal | present | absent |
| *Anolis chrysolepis* | absent | equal | present | between the joint |
| *Anolis fuscoauratus* | absent | equal | present | between the joint |
| *Anolis maculiventris* | absent | equal | present | between the joint |
| *Anolis mariarum* | absent | equal | present | absent/between the joint |
| *Anolis tolimensis* | absent | equal | present | surrounding the joint |
| *Anolis trachyderma* | absent | equal | present | between the joint |
| *Anolis ventrimaculatus* | absent | equal | present | absent between the joint |
| Hoplocercidae | | | | |
| *Hoplocercus spinosus* | absent | shorter | absent | between the joint |
| *Morunasaurus groi* | absent | shorter | present | absent |
| Tropiduridae | | | | |
| *Stenocercus erythrogaster* | absent | ? | absent | absent |
| *Stenocercus trachycephalus* | absent | equal | present | surrounding the joint |
| *Tropidurus pinima* | absent | shorter | present | absent |
| LACERTOIDEA | | | | |
| Gymnophthalmidae | | | | |
| *Anadia bogotensis* | absent | shorter | absent present | absent |
| *Gelanesaurus cochranae* | absent | shorter | absent | ? |
| *Loxopholis rugiceps* | absent | shorter | present | absent |

(Continued)

| Species | Columella | | | Joint of stapes |
|---|---|---|---|---|
| | Stapedial foramen | *Length of the columella | Widening of the osseous distal end | Connective tissue |
| *Neusticurus medemi* | absent | shorter | absent | absent |
| *Pholidobolus montium* | absent | shorter | absent | ? |
| *Pholidobolus vertebralis* | absent | shorter | present | absent |
| *Riama striata* | absent | equal | absent | surrounding the joint |
| *Tretioscincus bifasciatus* | absent | longer | present | surrounding the joint |
| Teiidae | | | | |
| *Cnemidophorus lemniscatus* | absent | ? | absent | absent |
| Lacertidae | | | | |
| *Acanthodactylus cf. schmidti* | absent | equal | present | surrounding the joint |
| SCINCOIDEA | | | | |
| Scincidae | | | | |
| *Mabuya falconensis* | absent | equal | present | absent |
| *Mabuya nigropunctatum* | absent | longer | present | between the joint |
| *Mabuya* sp. 1 | absent | equal | present | absent |
| *Mabuya* sp. 2 | absent | equal | present | between the joint |

**Note:**
(*) Length of the columella relative to that of the vertical axis of the extracolumella; (**?**) the condition of the specimen negated the ability to define this feature.

The files used in the analyses are available at Morphobank (*O'Leary & Kaufman, 2012*)—Project 3551 http://morphobank.org/permalink/?P3551.

## RESULTS

Lizards occupy a wide diversity of habitats (e.g., terrestrial, arboreal, saxicolous, fossorial, sand dwellers, semi-aquatic, and aquatic), and for this reason, it is expected that they exhibit significant variation in their middle ear structure depending on the way and medium through which they perceive sounds. As anticipated, according to the literature, the columella bone is a constant element with an uniaxial organization, although differs in shape and proportions (ranging from being long and thin as in *Tupinambis nigropunctatus* = *Tupinambis teguixin* (*Jollie, 1960*) to short and stumpy as in *Calyptommatus leiolepis* (*Holovacs et al., 2020*)). The extracolumella on the other hand, shows more significant variation in the number and shape of its processes (Fig. 1).

### Columella

The main body of the columella is an elongated osseous rod (Fig. 1A). Its proximal end is formed by an expanded footplate, which inserts into the oval window (the opening that leads to the inner ear); while at its distal end, the columella connects to the extracolumella. The variation found among the specimens examined was mainly in the presence of the stapedial foramen, the presence of a cartilaginous stalk on the distal end, differences in the length of the columella in relation to the extracolumellar vertical axis, and a slight

**Table 3 Characterization of the morphological variation of the extracolumella.**

| Species | Pars superior | Pars inferior | Anterior process | Posterior process | Internal process |
|---|---|---|---|---|---|
| GEKKOTA | | | | | |
| Gekkonidae | | | | | |
| *Hemidactylus brasilianus* | – posterior extension downward<br>– straight upper edge | thick with projections | long with small projections | short and pointed | absent |
| *Phelsuma madagascariensis* | – posterior extension downward<br>– straight upper edge | sharp | long with small projections | extended and thin | absent |
| Phyllodactylidae | | | | | |
| *Tarentola mauritanica* | – posterior extension downward<br>– straight upper edge | sharp | long with small projections | extended and thin | absent |
| *Thecadactylus rapicauda* | – posterior extension downward<br>– straight upper edge | thick with projections | long with small projections | extended and thin | absent |
| Pygopodidae | | | | | |
| *Lialis jicari* | – posterior extension straight<br>– straight upper edge | sharp | long pointed, downward | long and thick turned upward | absent |
| Sphaerodactylidae | | | | | |
| *Gonatodes albogularis* | – posterior extension downward<br>– straight upper edge | thick with projections | short, downward | short and pointed | absent |
| *Gonatodes concinnatus* | – posterior extension downward<br>– straight upper edge | thick with projections | short, downward | short and pointed | absent |
| IGUANIA | | | | | |
| Agamidae | | | | | |
| *Acanthocercus atricollis* | – no extension<br>– straight upper edge | sharp | long pointed and straight | extended and thin | present |
| *Leiolepis belliana* | – no extension<br>– straight upper edge | sharp | long pointed and straight | short and pointed | present |
| *Stellagama stellio* | – no extension<br>– rounded upper edge | sharp | absent | absent | present |
| Dactyloidae | | | | | |
| *Anolis antonii* | – no extension<br>– straight upper edge | sharp | short and pointed | short and pointed | present |
| *Anolis auratus* | – no extension<br>– straight upper edge | sharp | short and pointed | short and pointed | present |
| *Anolis chrysolepis* | – no extension<br>– straight upper edge | sharp | short and pointed | short and pointed | present |
| *Anolis fuscoauratus* | – no extension<br>– straight upper edge | sharp | short and pointed | short and pointed | present |
| *Anolis maculiventris* | – no extension<br>– straight upper edge | sharp | short and pointed | short and pointed | present |
| *Anolis mariarum* | – no extension<br>– straight upper edge | sharp | short and pointed | short and pointed | present |
| *Anolis tolimensis* | – no extension<br>– straight upper edge | sharp | short and pointed | short and pointed | present |
| *Anolis trachyderma* | – no extension<br>– straight upper edge | sharp | short and pointed | short and pointed | present |
| *Anolis ventrimaculatus* | – no extension<br>– straight upper edge | sharp | short and bifurcated | extended and thin | present |
| Hoplocercidae | | | | | |

(Continued)

| Species | Pars superior | Pars inferior | Anterior process | Posterior process | Internal process |
|---|---|---|---|---|---|
| *Hoplocercus spinosus* | – no extension<br>– rounded upper edge | sharp | long pointed and straight | extended and thin | present |
| *Morunasaurus groi* | – no extension<br>– rounded upper edge | sharp | long pointed and straight | short and pointed | present |
| Tropiduridae | | | | | |
| *Stenocercus erythrogaster* | – no extension<br>straight upper edge | sharp | long pointed and straight | extended and thin | present |
| *Stenocercus trachycephalus* | – no extension<br>– straight upper edge | sharp | long pointed and straight | extended and thin | present |
| *Tropidurus pinima* | – anterior extension straight<br> – straight upper edge | sharp | long pointed and straight | extended and thin | present |
| LACERTOIDEA | | | | | |
| Gymnophthalmidae | | | | | |
| *Anadia bogotensis* | – no extension<br>– straight upper edge | sharp | absent | short and pointed | absent |
| *Gelanesaurus cochranae* | – no extension<br>– straight upper edge | sharp | absent | short and pointed | absent |
| *Loxopholis rugiceps* | – no extension<br>– straight upper edge | sharp | absent | absent | absent |
| *Neusticurus medemi* | – no extension<br>– straight upper edge | sharp | absent | extended and thin | absent |
| *Pholidobolus montium* | – no extension<br>– straight upper edge | sharp | absent | short and pointed | absent |
| *Pholidobolus vertebralis* | – no extension<br>– straight upper edge | sharp | absent | absent | absent |
| *Riama striata* | – no extension<br>– straight upper edge | sharp | absent | short and pointed | absent |
| *Tretioscincus bifasciatus* | – no extension<br>– straight upper edge | sharp | absent | short and pointed | absent |
| Teiidae | | | | | |
| *Cnemidophorus lemniscatus* | – no extension<br>– straight upper edge | sharp | absent | absent | present |
| Lacertidae | | | | | |
| *Acanthodactylus* cf. *schmidti* | – no extension<br>– straight upper edge | sharp | long pointed and straight | short and pointed | present |
| SCINCOIDEA | | | | | |
| Scincidae | | | | | |
| *Mabuya falconensis* | – no extension<br>– tridentate upper edge | sharp | absent | absent | absent |
| *Mabuya nigropunctatum* | – no extension<br>– tridentate upper edge | sharp | absent | absent | absent |
| *Mabuya* sp. 1 | – no extension<br>– tridentate upper edge | sharp | absent | absent | absent |
| *Mabuya* sp. 2 | – no extension<br>– tridentate upper edge | sharp | absent | absent | absent |

**Table 4 Sources of the published data used to score the character states of the middle ear.**

| Group | Family | Species | References |
|---|---|---|---|
| Rhincocephalia | Sphenodontidae | *Sphenodon punctatus* | *Gray (1913)*, *Baird (1970)*, *Gans & Wever (1976)*, *Wever (1978)* |
| | Dibamidae | *Anelytropsis papillosus* | *McDowell (1967)*, *Greer (1976)*, *Wever (1978)* |
| Anguimorpha | Anguidae | *Anguis fragilis* | *Versluys (1898)*, *Wever (1973, 1978)* |
| | | *Anniella pulchra* | *Wever (1973, 1978)* |
| | | *Ophisaurus* | *Baird (1970)* |
| | Helodermatidae | *Heloderma suspectum* | *Versluys (1898)* |
| | Lanthanotidae | | *Wever (1978)* |
| | | *Lanthanotus borneensis* | *McDowell (1967)*, *Baird (1970)* |
| | Varanidae | *Varanus bengalensis* | *McDowell (1967)* |
| | | *Varanus niloticus* | *Versluys (1898)* |
| | | *Varanus salvator* | *Han & Young (2016)* |
| | Xenosauridae | *Xenosaurus grandis* | *Wever (1973, 1978)* |
| Gekkota | Eublepharidae | *Coleonyx variegatus* | *Posner & Chiasson (1966)* |
| | | *Eublepharis macularius* | *Wever (1978)*, *Werner et al. (2005, 2008)* |
| | Gekkonidae | *Chondrodactylus bibronii* (= *Pachydactylus bibronii*) | *Versluys (1898)* |
| | | *Gekko gecko* (= *Gecko verticillatus*) | *Versluys (1898)*, *Iordansky (1968)*, *Wever (1978)*, *Werner & Wever (1972)* |
| | | *Hemidactylus garnotti* | *Kluge & Eckhardt (1969)* |
| | | *Narudasia festiva* | *Daza, Aurich & Bauer (2012)* |
| | | *Uroplatus fimbriatus* | *Versluys (1898)* |
| | Pygopodidae | *Aprasia sps* | *Baird (1970)*, *Wever (1978)* |
| | | *Lialis burtonis* | *Wever (1974)* |
| | Sphaerodactylidae | *Teratoscincus scincus* | *Underwood (1957)*, *McDowell (1967)*, *Baird (1970)*, *Greer (1976)* |
| Iguania | Agamidae | *Bronchocela jubata* (= *Calotes jubatus*) | *Versluys (1898)* |
| | | *Ceratophora stoddarti* | *Wever (1973, 1978)* |
| | | *Ceratophora tennenti* | *Wever (1973, 1978)* |
| | | *Draco Volans* | *Versluys (1898)*, *Wever (1973, 1978)* |
| | | *Phrynocephalus maculatus* | *Wever (1973, 1978)* |
| | | *Phrynocephalus sp.* | *Wever (1973)* |
| | | *Uromastyx aegyptia* | *Versluys (1898)* |
| | Chamaleonidae | *Chamaeleo* | *Versluys (1898)*, *Wever (1968, 1978)* |
| | | *Rhampholeon* | *Toerien (1963)* |
| | Crotaphytidae | *Crotaphytus collaris* | *Wever & Werner (1970)*, *Wever (1978)* |
| | Iguanidae | *Iguana iguana* (= *Iguana tuberculata*) | *Versluys (1898)* |
| | Phrynosomatidae | *Callisaurus draconoides* | *Earle (1961c)*, *Wever (1973, 1978)* |
| | | *Cophosaurus texanus* | *Wever (1973, 1978)* |
| | | *Holbrookia* | *Earle (1961a, 1961c)*, *Baird (1970)* |
| | | *Holbrookia maculate* | *Earle (1961a, 1961c)*, *Wever (1973, 1978)* |
| | | *Phrynosoma coronatum* | *Wever (1973)* |

(Continued)

| Group | Family | Species | References |
|---|---|---|---|
| | | *Phrynosoma platyrhinos* | *Wever (1973, 1978)* |
| | | *Sceloporus magister* | *Wever (1967, 1973, 1978)* |
| Lacertoidea | Amphisbaenidae | *Amphisbaena* | *Gans & Wever (1972), Wever & Gans (1973), Olson (1966), Wever (1973)* |
| | | *Amphisbaena alba* | *Wever & Gans (1973)* |
| | | *Amphisbaena darwini trachura* | *Wever & Gans (1973)* |
| | | *Amphishenia manni* | *Wever & Gans (1973)* |
| | | *Amphisbaena fuliginosa* | *Versluys (1898)* |
| | | *Amphisbaena manni* | *Wever & Gans (1973)* |
| | | *Chirindia langi* | *Wever & Gans (1973)* |
| | | *Cynisca leucura* | *Wever & Gans (1973)* |
| | | *Monopeltis c. capensis* | *Wever & Gans (1973)* |
| | | *Zygaspis violacea* | *Wever & Gans (1973)* |
| | Bipedidae | *Bipes biporus* | *Wever & Gans (1972), Wever (1978)* |
| | Blanidae | *Blanus* | *Gans & Wever (1975), Wever (1978)* |
| | Lacertidae | *Podarcis muralis* (= *Lacerta muralis*) | *Wever (1978)* |
| | | *Timon lepidus* (= *Lacerta ocellata*) | *Versluys (1898)* |
| | Rhineuridae | *Rhineura floridana* | *Baird (1970), Olson (1966)* |
| | Teiidae | *Aspidoscelis tigris aethiops* (= *Cnemidophorus tessellatus aethiops*) | *Peterson (1966)* |
| | | *Pholidoscelis lineolatus* (= *Ameiva lineolata*) | *Wever (1978)* |
| | | *Tupinambis teguixin* (= *Tupinambis nigropunctatus*) | *Versluys (1898)* |
| | Trogonophidae | *Diplometopon zarudnyi* | *Gans & Wever (1975)* |
| | | *Trogonophis wiegmanni* | *Wever & Gans (1973)* |
| Scincoidea | Cordylidae | | *Wever (1978)* |
| | Gerrhosauridae | *Gerrhosaurus m. major* | *Wever (1978)* |
| | Scincidae | *Acontias plumbeus* | *Wever (1978)* |
| | | *Eutropis multifasciata* (= *Mabuia multifasciata*) | *Versluys (1898)* |
| | | *Feylinia currori* | *Greer (1976)* |
| | | *Feylinia polylepis* | *Greer (1976)* |
| | | *Scelotes bipes* | *Toerien (1963)* |
| | | *Trachylepis brevicollis* (= *Mabuya brevicollis*) | *Wever (1973, 1978)* |
| | Xantusiidae | *Lepidophyma gaigeae* | *Greer (1976), Wever (1978)* |
| | | *Lepidophyma flavimaculatum,* | *Wever (1978)* |
| | | *Lepidophyma smithi* | *Wever (1978)* |
| | | *Xantusia henshawi* | *Greer (1976), Wever (1978)* |
| | | *Xantusia riversiana* (= *Klauberrina riversiana*) | *Greer (1976)* |
| Serpentes | | | *Berman & Regal (1967), Wever (1978)* |

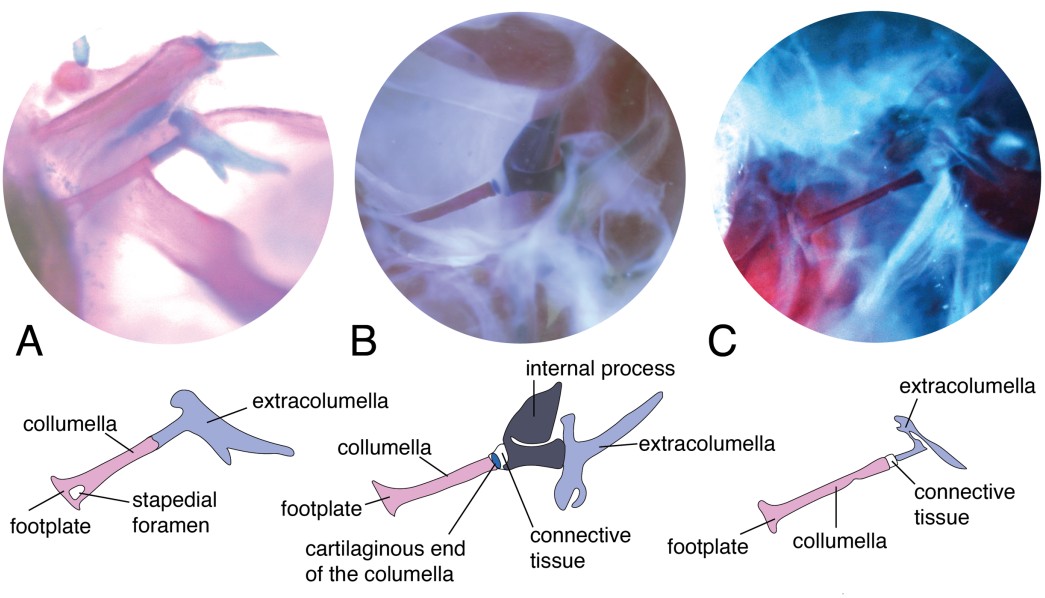

**Figure 2 Middle ear. The middle ear is shown from the posterior view of the skull.** The columella and the extracolumella (with its corresponding extracolumellar processes), have been sketched. (A) *Gonatodes concinnatus* MUJ 733; (B) *Hoplocercus* sp. MZUSP 92161; (C) *Tetrioscincus bifasciatus* ICN 5588. Scale bars: 1 mm.                               

expansion of the distal end. The variation of the columella observed in the examined specimens is summarized in Table 2.

The stapedial foramen (Fig. 2A) pierces the columella near the proximal end, and this opening allows the passage of the stapedial artery (*Greer, 1976*). In the present study, this character was observed in the gekkotans *Gonatodes albogularis*, *G. concinnatus*, *Hemidactylus brasilianus*, *Phelsuma madagascariensis*, and *Tarentola mauritanica* (Fig. 2A). This foramen is absent (Fig. 2B) in the remaining species studied, although it has been reported in lizards of the family Dibamidae (*Greer, 1976*; *Estes, de Queiroz & Gauthier, 1988*; *Gauthier, Estes & de Queiroz, 1988*) and embryonic stages of amphisbaenians (*Kearney, 2003*).

There are some differences in the relationship between the length of the columella and extracolumella. The length of the columella (measured from the footplate to the joint with the extracolumella; Fig. 1A), can be longer (Fig. 2C), subequal (Fig. 3A), or shorter (Fig. 3B), than the length of the extracolumellar vertical axis (taken from the upper edge of the pars superior to the lower edge of the pars inferior; Fig. 1B). In the specimens studied, the length of the columella was longer in *Acanthocercus atricollis* (Agamidae); *Mabuya nigropunctata* (Scincidae); *Tarentola mauritanica* (Phyllodactylidae); and *Tretioscincus bifasciatus* (Gymnophthalmidae; Fig. 2C). The columella length is similar to the extracolumella vertical axis in *Acanthodactylus* cf. *schmidti* (Lacertidae); *Anolis* spp., (Dactyloidae); *Hemidactylus brasilianus* and *Phelsuma madagascariensis* (Gekkonidae); *Mabuya* spp. (except in *M. nigropunctata*; Scincidae); *Riama striata* (Gymnophthalmidae); *Stenocercus trachycephalus* (Tropiduridae; Fig. 3A); and *Thecadactylus rapicauda*

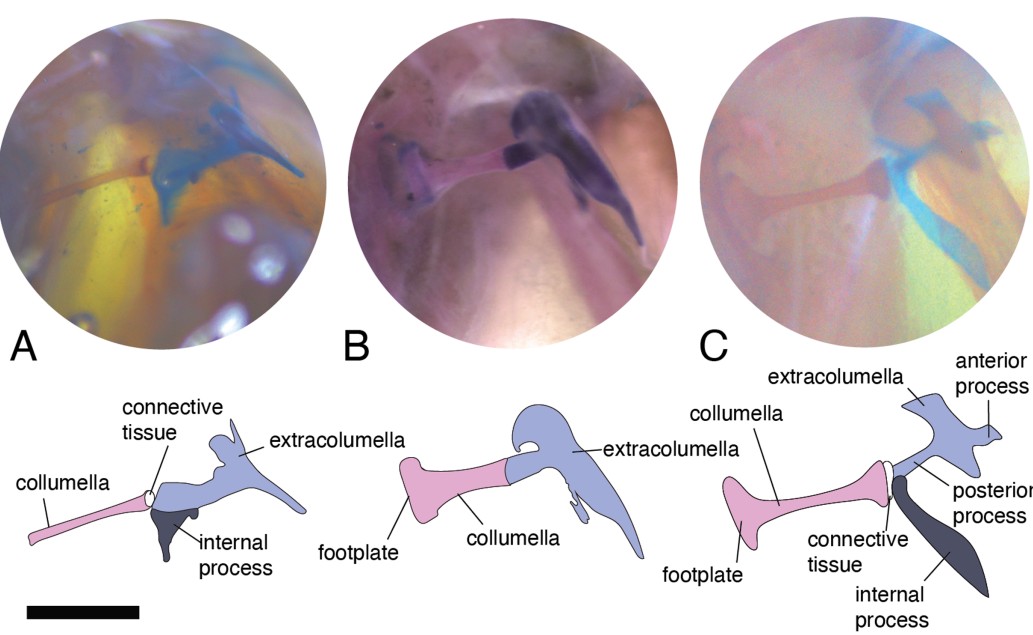

**Figure 3 Middle ear. The middle ear is shown from the posterior view of the skull. The columella and the extracolumella (with its corresponding extracolumellar processes), have been sketched.** (A) *Stenocercus trachycephalus* MUJ 635 (posterior view); (B) *Lialis jicari* MZUSP 67148 (posterior view); (C) *Anolis maculiventris* MHAU 10468 (posterior view). Scale bars: 2 mm.

(Phyllodactylidae). The columella was shorter in *Anadia bogotensis, Gelanesaurus cochranae, Loxopholis rugiceps, Neusticurus medemi, Pholidobolus montium*, and *P. vertebralis* (Gymnophthalmidae); *Gonatodes albogularis, G. concinnatus* (Sphaerodactylidae; Fig. 2A); *Hoplocercus spinosus, Morunasaurus groi* (Hoplocercidae); *Lialis jicari* (Pygopodidae; Fig. 3B); and *Tropidurus pinima* (Tropiduridae).

A slight expansion of the osseous distal end of the columella was observed in *Acanthocercus atricollis* (Agamidae); *Acanthodactylus* cf. *schmidti* (Lacertidae); *Anolis* spp. (Dactyloidae; Fig. 3C); *Mabuya* spp. (Scincidae); *Morunasaurus groi* (Hoplocercidae); *Lialis jicari* (Pygopodidae; Fig. 3B); *Loxopholis rugiceps, Pholidobolus vertebralis, Tretioscincus bifasciatus* (Gymnophthalmidae; Fig. 2C); *Phelsuma madagascariensis* (Gekkonidae); *Stenocercus trachycephalus* (Fig. 3A); and *Tropidurus pinima* (Tropiduridae). The remaining species do not show this expansion. Two conditions of the distal end of the columella—expanded end or constant size along the columellar shaft—were observed in different specimens of *Anadia bogotensis* (Gymnophthalmidae), specimen ICN 2987 (slight expansion) and ICN 2178 (constant width).

We detected a slight difference in the cartilaginous rim of the footplate. The rim can form a complete ring around the footplate of the columella, as observed in *Gonatodes albogularis* MUJ-665, or be a discontinuous and very thin ring, as observed in *Anolis auratus* MUJ 590. In some specimens, this ring is absent altogether (e.g., *Pholidobolus vertebralis* ICN 5719). We do not discount that differences in the development of the

cartilaginous ring of the footplate could be an artifact of the staining used in the preparations, and may not represent true morphological variation.

## Columella–extracolumella joint

This joint varies in the presence/absence of connective tissue and the form of the joint. Connective tissue was observed in *Acanthocercus atricollis* (Agamidae); *Acanthodactylus* cf. *schmidti* (Lacertidae); *Anolis* spp., except *A. auratus* (Dactyloidae; Fig. 3C); *Hoplocercus spinosus* (Hoplocercidae; Fig. 2B); *Mabuya nigropunctata, Mabuya* sp. 2 (Scincidae); *Riama striata, Tretioscincus bifasciatus* (Gymnophthalmidae; Fig. 2C); *Stenocercus trachycephalus* (Tropiduridae; Fig. 3A); and *Tarentola mauritanica* (Phyllodactylidae). When the two elements are joined by connective tissue, the lateral end of the columella is cartilaginous. This condition was observed in *Anolis antonii, A. chrysolepis, A. fuscoauratus, A. maculiventris, A. trachyderma* (Dactyloidae); *Hoplocercus spinosus* (Hoplocercidae; Fig. 2B); *Mabuya nigropunctata* and *Mabuya* sp. 2 (Scincidae). When the connective tissue is surrounding the columella–extracolumella joint, the cartilaginous shaft of the columella is hidden. This formation of joint and connective tissue was observed in *Acanthocercus atricollis* (Agamidae); *Acanthodactylus* cf. *schmidti* (Lacertidae); *Anolis tolimensis* (Dactyloidae); *Riama striata, Tretioscincus bifasciatus* (Gymnophthalmidae; Fig. 2C); *Stenocercus trachycephalus* (Tropiduridae); and *Tarentola mauritanica* (Phyllodactylidae). The remaining specimens do not show connective tissue (Figs. 2A and 3B). The specimens of *Anolis mariarum* and *A. ventrimaculatus* exhibit variation in the presence of the connective tissue. In specimens ICN 5808 and MHUA 10014 of *A. mariarum* the connective tissue is seen between the joint, while specimen MHUA 10013 does not have connective tissue; and in *A. ventrimaculatus*, the specimens MHUA 10671 and MHUA 10672 display the connective tissue between the joint, while in specimen PUJ 338 connective tissue is absent.

## Extracolumella

Usually, this element is cartilaginous, and composed of a small shaft, two to four processes attached to the tympanic membrane, and the internal process (Fig. 1B) which is present only in iguanians and related species. The extracolumella was present in all the specimens examined and exhibits large morphological disparity among lizards.
The variation in this element involves the presence/absence of the anterior and/or posterior process, the shape of the four processes, and the presence/absence of the internal process. The extracolumella variation observed in the examined specimens is summarized in Table 3.

In the specimens studied, the extracolumella exhibits four processes – superior and inferior pars, and the anterior and posterior processes – all attached to the tympanic membrane (Fig. 1B). The pars superior and the pars inferior form the vertical axis of the extracolumella, and from this axis, the anterior and posterior processes arise laterally. The variation observed in this pattern is the lack of the anterior process in some species, or the lack of both processes (anterior and posterior) in others. The general pattern (the presence of four processes of the extracolumella; Fig. 1B), was observed in the specimens of

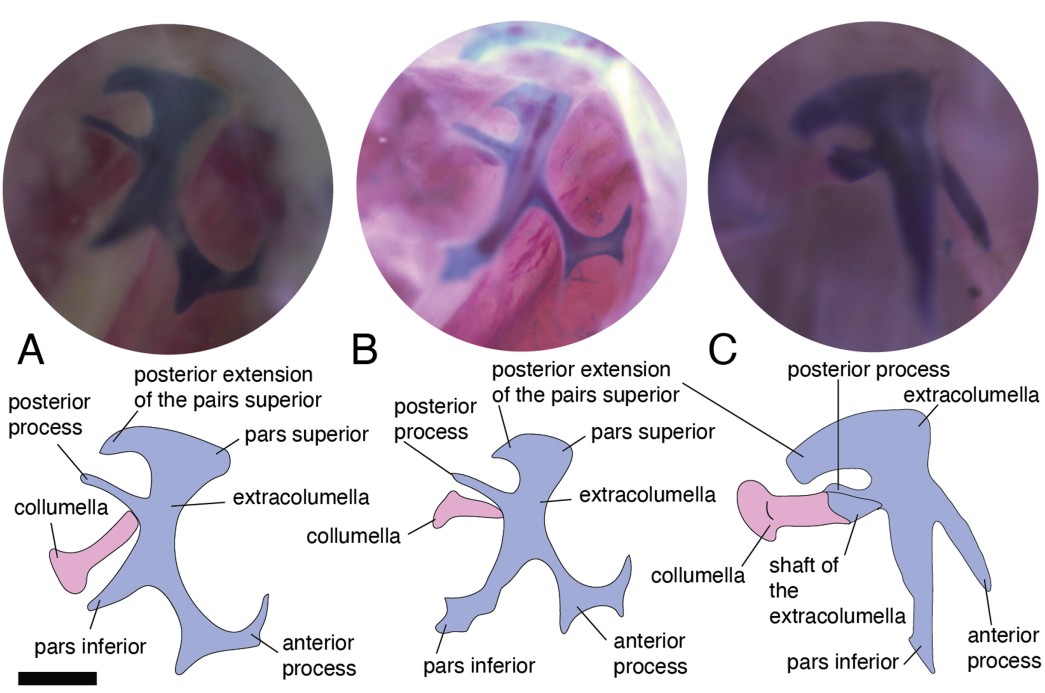

**Figure 4 Extracolumella.** The extracolumella is shown from the lateral view of the skull. The columella and the extracolumella (with its corresponding extracolumellar processes), have been sketched. (A) *Phelsuma madagascariensis* MZUSP 36938; (B) *Thecadactylus rapicauda* MZUSP 97833; (C) *Lialis jicari* MZUSP 67148. Scale bars: 1 mm.    

*Acanthocercus atricollis, Leiolepis belliana* (Agamidae); *Acanthodactylus* cf. *schmidti* (Lacertidae); *Anolis* spp. (Dactyloidae); *Hemidactylus brasilianus, Phelsuma madagascariensis* (Gekkonidae; Fig. 4A); *Tarentola mauritanica, Thecadactylus rapicauda* (Phyllodactylidae; Fig. 4B); *Lialis jicari* (Pygopodidae; Fig. 4C); *Gonatodes albogularis, G. concinnatus* (Sphaerodactylidae; Fig. 5A); *Hoplocercus spinosus, Morunasaurus groi* (Hoplocercidae; Fig. 5B); *Stenocercus erythrogaster, S. trachycephalus*, and *Tropidurus pinima* (Tropiduridae; Fig. 5C). The anterior process is absent in *Anadia bogotensis, Gelanesaurus cochranae* (Fig. 6A), *Loxopholis rugiceps, Neusticurus medemi, Pholidobolus montium, P. vertebralis, Riama striata, Tretioscincus bifasciatus* (Gymnophthalmidae); *Stellagama stellio* (Agamidae; Fig. 6B); *Cnemidophorus lemniscatus* (Teiidae); and *Mabuya* spp. (Scincidae; Fig. 6C). The posterior process is absent in *Cnemidophorus lemniscatus* (Teiidae); *Loxopholis rugiceps, Pholidobolus vertebralis* (Gymnophthalmidae); *Mabuya* spp. (Scincidae; Fig. 6C); and *Stellagama stellio* (Agamidae; Fig. 6B).

All four extracolumellar processes display some morphological variation in their shape. The pars superior shows two principal variations, determined by the presence of an extension of the upper edge, which varies in the orientation of the extension (anterior or posterior). The upper edge of the pars superior has one posterior extension in the gekkotans *Gonatodes albogularis, G. concinnatus* (Fig. 5A), *Hemidactylus brasilianus, Lialis jicari* (Fig. 4C), *Phelsuma madagascariensis* (Fig. 4A), *Tarentola mauritanica*, and *Thecadactylus rapicauda* (Fig. 4B); while in *Tropidurus pinima* (Tropiduridae), the extension is anterior (Fig. 5C). In all of these species, the distal end of the posterior

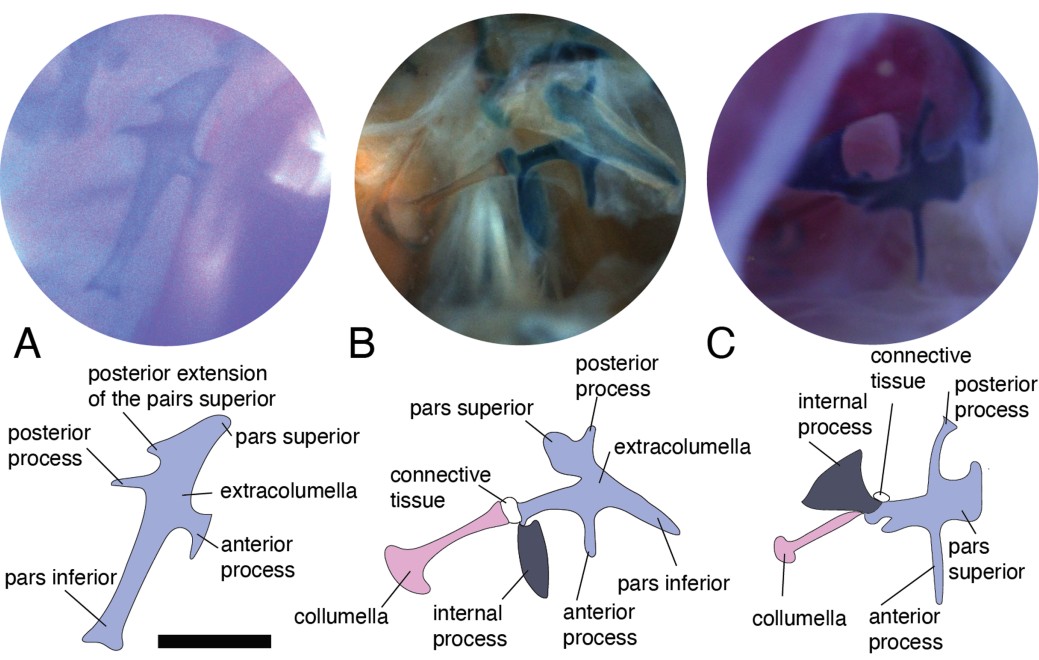

**Figure 5 Extracolumella.** The extracolumella is shown from different views of the skull. The columella and the extracolumella (with its corresponding extracolumellar processes), have been sketched. (A) *Gonatodes concinnatus* MUJ 733 (from the lateral view of the skull); (B) *Morunasaurus groi* ICN 6270 (from the posterior view of the skull); (C) *Tropidurus pinima* MZUSP 92140 (from the ventrolateral view of the skull). Scale bars: 1 mm.

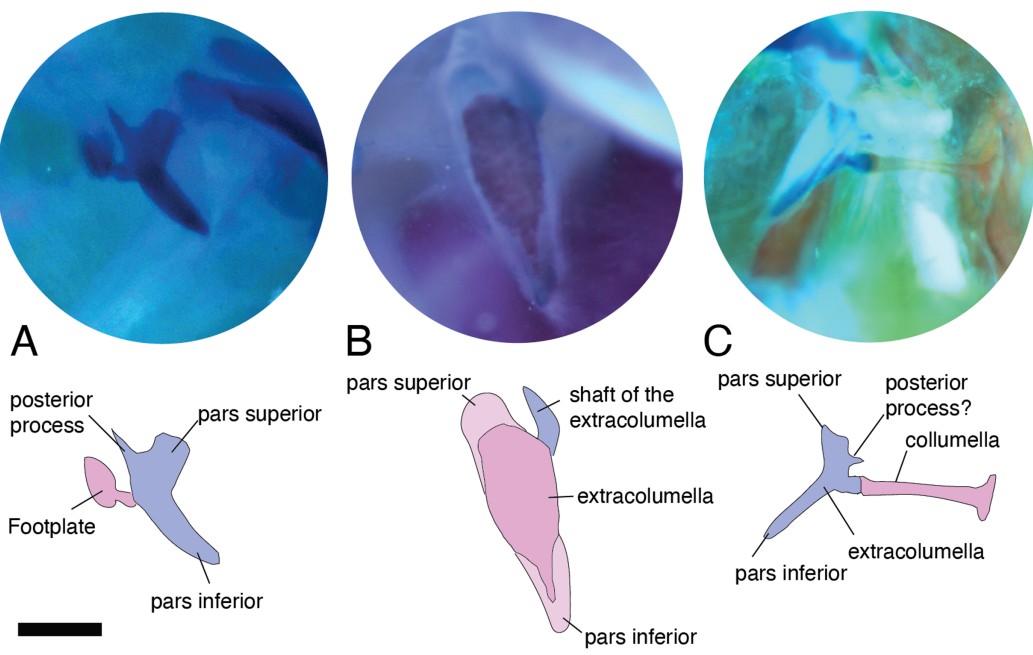

**Figure 6 Extracolumella.** The extracolumella is shown from different views of the skull. The columella and the extracolumella (with its corresponding extracolumellar processes), have been sketched. (A) *Gelanosaurus cochrane* ICN 9453 (from the lateral view of the skull); (B) *Stellagama stellio* MZUSP 95176 (from the lateral view of the skull); (C) *Mabuya falconensis* ICN 11312 (from the posterior view of the skull). Scale bars: 1 mm.

extension of the pars superior is curved downward, except in *Lialis jicari* (Fig. 4C) in which this distal end is slightly straight, like the anterior extension in *Tropidurus pinima* (Fig. 5C). The remaining species lack any of these extensions. Additionally, the upper edge of the pars superior displays three kinds of surfaces: a slightly plane edge (Figs. 4A–4C, 5A, 5C and 6A), a rounded edge (Figs. 5B and 6B), and an edge with small tubercles (Fig. 6C). The upper edge is slightly plane in *Acanthocercus atricollis, Leiolepis belliana* (Agamidae); *Acanthodactylus* cf. *schmidti* (Lacertidae); *Anadia bogotensis, Gelanesaurus cochranae* (Fig. 6A), *Loxopholis rugiceps, Neusticurus medemi, Pholidobolus montium, P. vertebralis, Riama striata, Tretioscincus bifasciatus* (Gymnophthalmidae); *Anolis* spp. (Dactyloidae); *Cnemidophorus lemniscatus* (Teiidae); *Gonatodes albogularis, G. concinnatus* (Sphaerodactylidae; Fig. 5A); *Hemidactylus brasilianus, Phelsuma madagascariensis* (Gekkonidae; Fig. 4A); *Lialis jicari* (Pygopodidae; Fig. 4C); *Stenocercus erythrogaster, S. trachycephalus, Tropidurus pinima* (Tropiduridae; Fig. 5C); *Tarentola mauritanica* and *Thecadactylus rapicauda* (Phyllodactylidae; Fig. 4B); while the edge is rounded in *Stellagama stellio* (Agamidae; Fig. 6B); *Hoplocercus spinosus* and *Morunasaurus groi* (Hoplocercidae; Fig. 5B). Finally, an edge with three small tubercles is observed in the specimens of *Mabuya* spp. (Scincidae; Fig. 6C).

The pars inferior is the extracolumellar process with the most conservative morphology. This process displays an inverted triangular shape, with the thicker portion contacting the pars superior (Fig. 1B), and the thinner portion at the distal end. The only variation observed is in the distal end which can appear sharp or thick. The sharp distal end (Fig. 1B) is present in all the specimens studied except in *Gonatodes albogularis, G. concinnatus* (Sphaerodactylidae; Fig. 5A); *Hemidactylus brasilianus* (Gekkonidae); and *Thecadactylus rapicauda* (Phyllodactylidae; Fig. 4B) which shows a thick distal end with small projections on the pars inferior.

Both processes, anterior and posterior, arise from the superior half of the vertical axis of the extracolumella, which is formed by the pars superior and inferior (Fig. 1B). Usually, the processes are thin and extended laterally, but in some species, these are thick and/or turned downward (see below). The anterior process appears in three main shapes: short (Fig. 3C), long and pointed (Figs. 4C and 5B–5C), or long with some small and sharp projections (Figs. 4A and 4B). The first type, a short and pointed anterior process, is the simplest morphology for this process and was observed in the studied specimens of *Anolis* spp. (Fig. 3C), except *A. ventrimaculatus* (Dactyloidae) which shows a short process, but its distal end has two small pointed prolongations (see below). The second type, a long and pointed process, was observed in *Acanthocercus atricollis, Leiolepis belliana* (Agamidae); *Acanthodactylus* cf. *schmidti* (Lacertidae); *Hoplocercus spinosus, Morunasaurus groi* (Hoplocercidae; Fig. 5B); *Lialis jicari* (Pygopodidae; Fig. 4C); *Stenocercus erythrogaster, S. trachycephalus*, and *Tropidurus pinima* (Tropiduridae; Fig. 5C). In *Lialis jicari* (Fig. 4C), the anterior process is oriented downward, while in the other species this process is straight. The third type, a long thick extension with some small and sharp prolongations (Figs. 4A and 4B) was observed in *Hemidactylus brasilianus* and *Phelsuma madagascariensis* (Gekkonidae; Fig. 4A); *Tarentola mauritanica* and *Thecadactylus rapicauda* (Phyllodactylidae; Fig. 4B). Unlike the previous species,

*Gonatodes albogularis* and *G. concinnatus* (Sphaerodactylidae; Fig. 5A) present short anterior processes with the distal ends turning downward, simulating a hook that is rounded in *G. albogularis*, while it forms a right angle in *G. concinnatus* (Fig. 5A). There is no anterior process in the specimens of *Anadia bogotensis, Gelanesaurus cochranae* (Fig. 6A), *Loxopholis rugiceps, Nesticurus medemi, Riama striata, Tretioscincus bifasciatus* (Gymnophthalmidae); *Cnemidophorus lemniscatus* (Teiidae); all specimens of *Mabuya* spp. (Scincidae; Fig. 6C); or *Stellagama stellio* (Agamidae; Fig. 6B).

The posterior process shows a slight variation in both the length and thickness of its extension. Among the specimens studied, most of them show an extended and thin, or a short and acute process, except for *Lialis jicari* (Pygopodidae) which shows a short thick posterior process turned upward, simulating a hook (Fig. 4C). The extended thin posterior process was observed in *Acanthocercus atricollis* (Agamidae); *Anolis ventrimaculatus* (Dactyloidae); *Hoplocercus spinosus* (Hoplocercidae); *Nesticurus medemi* (Gymnophthalmidae); *Phelsuma madagascariensis* (Gekkonidae; Fig. 4A); *Stenocercus erythrogaster, S. trachycephalus, Tropidurus pinima* (Tropiduridae; Fig. 5C); *Tarentola mauritanica* and *Thecadactylus rapicauda* (Phyllodactylidae; Fig. 4B); while the short and acute posterior process was observed in *Acanthodactylus* cf. *schmidti* (Lacertidae); *Anadia bogotensis, Gelenasaurus cochranae* (Fig. 6A), *Pholidobolus montium, Riama striata, Tretioscincus bifasciatus* (Gymnophthalmidae); *Anolis* spp., except *A. ventrimaculatus* (Dactyloidae); *Gonatodes albogularis, G. concinnatus* (Sphaerodactylidae; Fig. 5A); *Hemidactylus brasilianus* (Gekkonidae); *Leiolepis belliana* (Agamidae); and *Morunasaurus groi* (Hoplocercidae; Fig. 5B). The specimens of *Cnemidophorus lemniscatus* (Teiidae); *Loxopholis rugiceps* (Gymnophthalmidae); *Mabuya* spp. (Scincidae; Fig. 6C); and *Stellagama stellio* (Agamidae; Fig. 6B) do not show the posterior process.

In some specimens, the extracolumella, usually cartilaginous, exhibits a red-stained region of different sizes and in different degrees of staining, in the central axis, and the lateral processes, indicating the presence of osseous tissue. This feature was observed in *Acanthocercus atricollis, Leiolepis belliana, Stellagama stellio* (Agamidae; Fig. 6B); *Anolis* spp. (Dactyloidae; Fig. 3C); *Hemidactylus brasilianus* (Gekkonidae); *Morunasaurus groi* (Hoplocercidae; Fig. 5B); *Stenocercus trachycephalus* (Tropiduridae; Fig. 3A); and *Thecadactylus rapicauda* (Phyllodactylidae; Fig. 4B). This feature is particularly noticeable in some specimens of the *Anolis* species in which the red-stained area appears bigger and more intense than in the other species.

## Internal process

This process originates from the shaft of the extracolumella and extends laterally to contact the tympanic conch of the quadrate bone. It is fan-shaped, with a thin origin at the shaft of the extracolumella, but expands distally to develop a broad edge. This process was only found in *Acanthocercus atricollis, Leiolepis belliana, Stellagama stellio* (Agamidae); *Acanthodactylus* cf. *schmidti* (Lacertidae); *Anolis* spp. (Dactyloidae; Fig. 3C); *Cnemidophorus lemniscatus* (Teiidae); *Hoplocercus spinosus, Morunasaurus groi* (Hoplocercidae; Fig. 5B); and *Stenocercus erythrogaster, S. trachycephalus*, and *Tropidurus pinima* (Tropiduridae; Fig. 5C). This process is absent in the remaining studied species.

The internal process varies in the width of the origin at the junction with the extracolumella. The internal process is triangular with a thin origin and a very differentiated distal edge in *L. belliana, A.* cf. *schmidti,* and *T. pinima* (Fig. 5C), while in other species, the origin is broad (e.g., *A. atricollis, S. stellio; Anolis* spp., *H. spinosus, M. groi;* and *S. trachycephalus*). Although in the specimens studied of *C. lemniscatus* and *S. erythrogaster,* the internal process was evident, it was not possible to determine the size of its origin due to mechanical damage caused by inadequate specimen preparation.

## Ancestral reconstruction

### Definition of characters

Based on the morphological descriptions presented above, the following middle ear characters were defined to analyze them in a phylogenetic framework. Despite the limited sampling, the results of this survey provide a baseline to understand overall variation and outline a general scenario about the evolutionary changes of selected features of the middle ear in lizards.

– Character 1. Length of the columella relative to the extracolumella central axis length. [0] equal length (Fig. 2C); [1] longer (Fig. 3A); and [2] shorter (Fig. 3B).
– Character 2. Extracolumella. [0] simple (Fig. 4A); [1] complex; [2] elongated; [3] absent. To test if there is a general pattern in the reduction of the extracolumellar processes, we summarized the available information on this structure into four states, including the absence of the extracolumella. The state [0] refers to the extracolumellas that have at least three processes regardless of the size of each, while the state [1] indicates the extracolumella with the four developed processes—the superior and inferior pars, and the anterior and posterior processes. Finally, an elongated extracolumella refers to a case where this structure runs anteriorly along the quadrate and mandible and contacts the skin; this kind of extracolumella does not show any processes.
– Character 3. Nature of the Internal Process. [0] Absent (Fig. 2A); [1] present (Fig. 2B).

### Character mapping

Characters were optimized using parsimony with unordered states and equal weights, and Bayesian analyses with the all rates different (ARD) and the equal rates (ER) models. The summaries of the optimization of characters with parsimony are presented in Figs. 7 and 8, and the values of the posterior probabilities of the Bayesian reconstructions in Table 5. The complete mapping with parsimony (Fig. S1) and Bayesian reconstructions (Figs. S2–S4), and the posterior probability values (Table S3) are also available on Morphobank (*O'Leary & Kaufman, 2012*)—Project 3551 http://morphobank.org/permalink/?P3551.

### Character 1. Length of the columella relative to the extracolumella central axis length

The parsimony approach (Fig. 7; Fig. S1) shows the ancestral condition of the columella's length relative to the extracolumella central axis length for Squamata [node 2] as
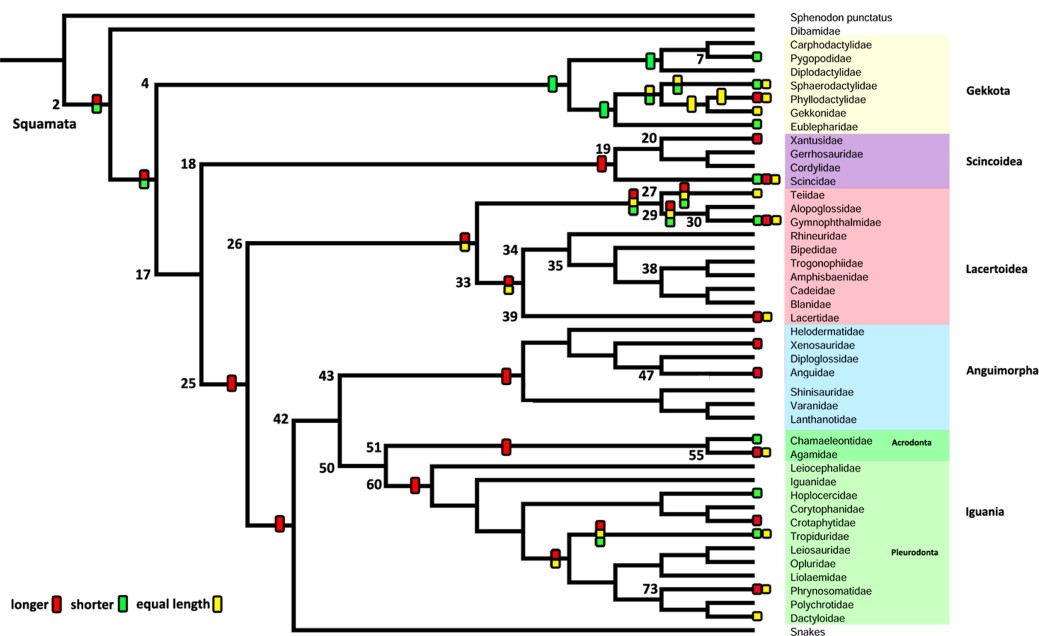

**Figure 7 Summary of the mapping of the characters using maximum parsimony (MP).** Character 1. Length of the columella relative to the extracolumella central axis length.

ambiguous between the states shorter and longer. Also, there is ambiguity between the three states of the character for the ancestor of Teiioidea [27], and between the states longer and equal length in Lacertoidea [26], Lacertidae [39], (Amphisbaenidae + Lacertidae) [33]. The shorter columella state was the reconstructed state for the ancestral node of Gekkota [4] and Pygopodidae [7]; and the longer columella state for the nodes of (Xantusiidae (Gerrhosauridae + Cordylidae)) [19], Scincoidea [18], Anguimorpha [43], Agamidae [55], Acrodonta [51], Phrynosomatidae [73], Pleurodonta [60], Iguania [50] (Anguimorpha + Iguania) [42], (Lacertoidea (Serpentes (Anguimorpha + Iguania))) [25], (Scincoidea (Lacertoidea (Serpentes (Anguimorpha + Iguania)))) [17]. There is no available information for the clades Amphisbaenia [34], (Amphisbaenidae + Trogonophidae) [38], in (Bipedidae ((Cadeidae + Blanidae) (Amphisbaenidae + Trogonophidae))) [35].

The Bayesian analysis (Table 5; Fig. S2; Table S3) with both models shows ambiguity for the ancestral node of Squamata [2] with equal probabilities for all states. The ARD reconstruction found ambiguity for all other clades with similar values for each state. However, the higher support values for these clades are for the longer columella. Similarly, the ER reconstruction found ambiguity for all clades with equal values of probability for each character state for all these clades.

### Character 2. Extracolumella

The parsimony approach (Fig. 8A, Fig. S1) defines the simple extracolumella as the ancestral condition for Squamata [node 2]. This state was also reconstructed for the nodes of the clades Scincoidea [18], Teiioidea [27], Lacertidae [39], Amphisbaenia [34],
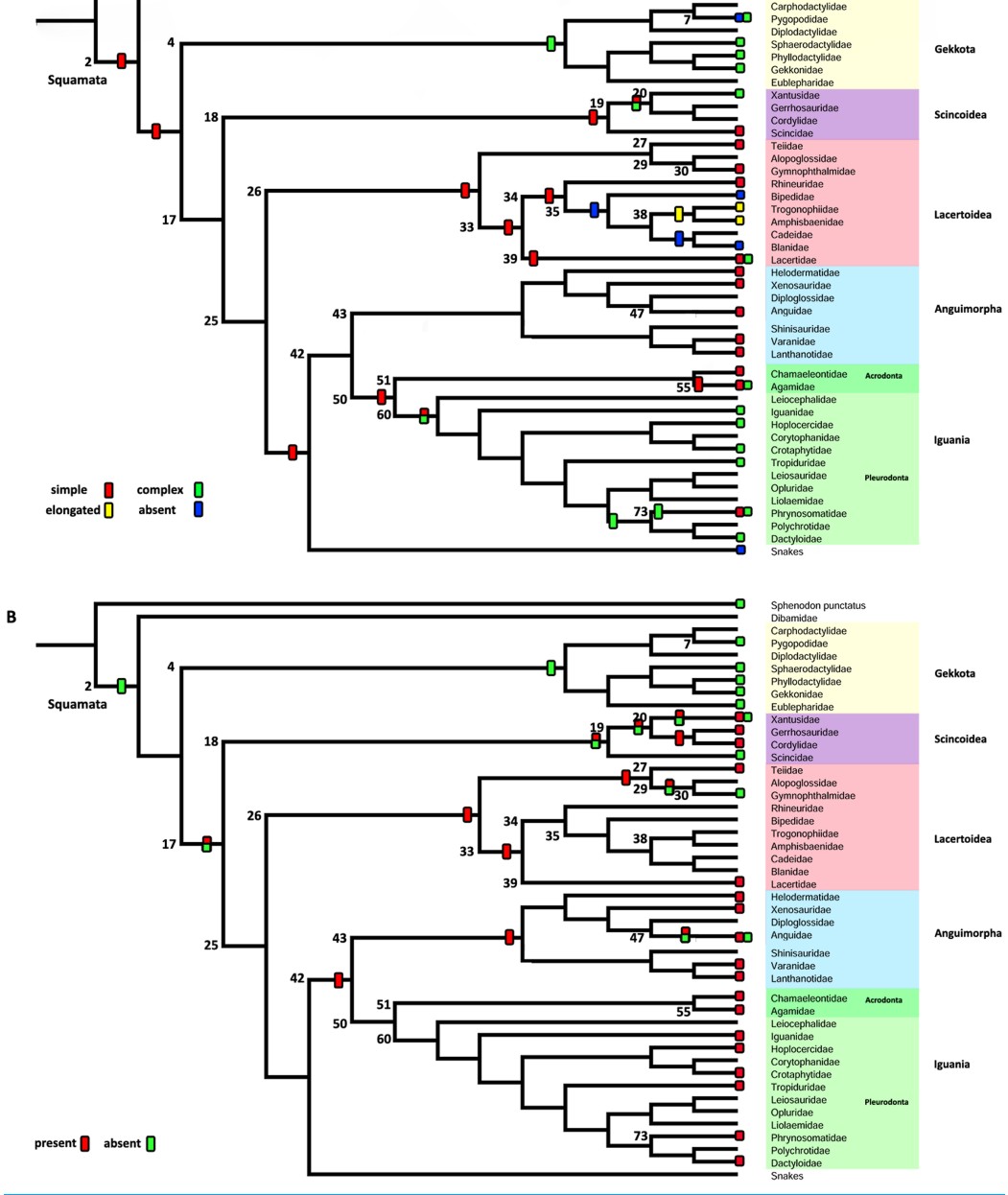

**Figure 8 Summary of the mapping of the characters using maximum parsimony (MP).** (A) Character 2. Extracolumella. (B) Character 3. Internal Process.

(Amphisbaenidae + Lacertidae) [33], Lacertoidea [26], Anguimorpha [43], Agamidae [55], Acrodonta [51], Iguania [50], (Anguimorpha + Iguania) [42], and (Lacertoidea (Serpentes (Anguimorpha + Iguania))) [25], (Scincoidea (Lacertoidea (Serpentes (Anguimorpha + Iguania)))) [17]. The complex extracolumella was the estimated ancestral state in Gekkota [4], Pygopodidae [7], and Phrynosomatidae [73]; the elongated extracolumella in (Amphisbaenidae + Trogonophidae) [38]; and the absence of extracolumella in (Bipedidae ((Cadeidae + Blanidae) (Amphisbaenidae + Trogonophidae))) [35]. This reconstruction

**Table 5 Summary of the posterior probabilities estimated for each node by the Bayesian ancestral state reconstructions modelled using the models with all rates different (ARD) and equal rates (ER).**

| Node | Character 1 ARD model | | | | Character 1 ER model | | | |
|---|---|---|---|---|---|---|---|---|
| | – | Equal | Longer | Shorter | – | Equal | Longer | Shorter |
| 2 | 0.13 | 0.31 | **0.33** | 0.23 | 0.25 | 0.25 | 0.25 | 0.25 |
| 4 | 0.13 | 0.31 | **0.33** | 0.23 | 0.25 | 0.25 | 0.25 | 0.25 |
| 7 | 0.13 | 0.31 | **0.33** | 0.23 | 0.25 | 0.25 | 0.25 | 0.25 |
| 17 | 0.14 | 0.31 | **0.32** | 0.23 | 0.25 | 0.25 | 0.25 | 0.25 |
| 18 | 0.13 | 0.31 | **0.33** | 0.23 | 0.25 | 0.25 | 0.25 | 0.25 |
| 19 | 0.13 | 0.31 | **0.33** | 0.23 | 0.25 | 0.25 | 0.25 | 0.25 |
| 20 | 0.13 | 0.31 | **0.33** | 0.23 | 0.25 | 0.25 | 0.25 | 0.25 |
| 25 | 0.13 | 0.31 | **0.33** | 0.23 | 0.25 | 0.25 | 0.25 | 0.25 |
| 26 | 0.13 | 0.31 | **0.33** | 0.23 | 0.25 | 0.25 | 0.25 | 0.25 |
| 27 | 0.13 | 0.31 | **0.33** | 0.23 | 0.25 | 0.25 | 0.25 | 0.25 |
| 29 | 0.14 | 0.31 | **0.32** | 0.23 | 0.25 | 0.25 | 0.25 | 0.25 |
| 30 | 0.13 | 0.31 | **0.33** | 0.23 | 0.25 | 0.25 | 0.25 | 0.25 |
| 33 | 0.13 | 0.31 | **0.33** | 0.23 | 0.25 | 0.25 | 0.25 | 0.25 |
| 34 | 0.13 | 0.31 | **0.33** | 0.23 | 0.25 | 0.25 | 0.25 | 0.25 |
| 35 | 0.13 | 0.31 | **0.33** | 0.23 | 0.25 | 0.25 | 0.25 | 0.25 |
| 38 | 0.13 | 0.31 | **0.33** | 0.23 | 0.25 | 0.25 | 0.25 | 0.25 |
| 39 | 0.13 | 0.31 | **0.33** | 0.23 | 0.25 | 0.25 | 0.25 | 0.25 |
| 42 | 0.15 | 0.30 | **0.31** | 0.24 | 0.25 | 0.25 | 0.25 | 0.25 |
| 43 | 0.13 | 0.31 | **0.33** | 0.23 | 0.25 | 0.25 | 0.25 | 0.25 |
| 47 | 0.13 | 0.31 | **0.33** | 0.23 | 0.25 | 0.25 | 0.25 | 0.25 |
| 50 | 0.13 | 0.31 | **0.33** | 0.23 | 0.25 | 0.25 | 0.25 | 0.25 |
| 51 | 0.13 | 0.31 | **0.33** | 0.23 | 0.25 | 0.25 | 0.25 | 0.25 |
| 55 | 0.13 | 0.31 | **0.33** | 0.23 | 0.25 | 0.25 | 0.25 | 0.25 |
| 60 | 0.13 | 0.31 | **0.33** | 0.23 | 0.25 | 0.25 | 0.25 | 0.25 |
| 73 | 0.13 | 0.31 | **0.33** | 0.23 | 0.25 | 0.25 | 0.25 | 0.25 |

| Node | Character 2 ARD model | | | | Character 2 ER model | | | |
|---|---|---|---|---|---|---|---|---|
| | Absent | Expanded | Extensive | Reduced | Absent | Expanded | Extensive | Reduced |
| 2 | 0.01 | 0.04 | 0.00 | **0.95** | 0.00 | 0.07 | 0.00 | **0.93** |
| 4 | 0.15 | **0.82** | 0.01 | 0.02 | 0.00 | **0.99** | 0.00 | 0.01 |
| 7 | 0.14 | **0.85** | 0.01 | 0.00 | 0.09 | **0.91** | 0.00 | 0.00 |
| 17 | 0.00 | 0.01 | 0.00 | **0.99** | 0.00 | 0.04 | 0.00 | **0.96** |
| 18 | 0.01 | 0.03 | 0.00 | **0.96** | 0.00 | 0.06 | 0.00 | **0.94** |
| 19 | 0.03 | 0.12 | 0.01 | **0.84** | 0.01 | 0.16 | 0.01 | **0.82** |
| 20 | 0.14 | **0.80** | 0.02 | 0.04 | 0.00 | **0.98** | 0.00 | 0.02 |
| 25 | 0.00 | 0.00 | 0.00 | **1.00** | 0.00 | 0.00 | 0.00 | **1.00** |
| 26 | 0.00 | 0.00 | 0.00 | **1.00** | 0.00 | 0.00 | 0.00 | **1.00** |
| 27 | 0.00 | 0.00 | 0.00 | **1.00** | 0.00 | 0.00 | 0.00 | **1.00** |
| 29 | 0.00 | 0.00 | 0.00 | **1.00** | 0.00 | 0.00 | 0.00 | **1.00** |
| 30 | 0.00 | 0.00 | 0.00 | **1.00** | 0.00 | 0.00 | 0.00 | **1.00** |
| 33 | 0.00 | 0.00 | 0.00 | **1.00** | 0.00 | 0.00 | 0.00 | **1.00** |

*(Continued)*

## Table 5 (continued)

| Node | Character 2 ARD model | | | | Character 2 ER model | | | |
|------|--------|----------|-----------|---------|--------|----------|-----------|---------|
| | Absent | Expanded | Extensive | Reduced | Absent | Expanded | Extensive | Reduced |
| 34 | 0.00 | 0.00 | 0.01 | **0.99** | 0.02 | 0.00 | 0.01 | **0.97** |
| 35 | 0.00 | 0.00 | **0.94** | 0.06 | **0.58** | 0.00 | 0.33 | 0.09 |
| 38 | 0.00 | 0.00 | **0.96** | 0.04 | 0.30 | 0.00 | **0.70** | 0.00 |
| 39 | 0.01 | 0.07 | 0.00 | **0.92** | 0.00 | 0.08 | 0.00 | **0.92** |
| 42 | 0.00 | 0.00 | 0.00 | **1.00** | 0.00 | 0.00 | 0.00 | **1.00** |
| 43 | 0.00 | 0.00 | 0.00 | **1.00** | 0.00 | 0.00 | 0.00 | **1.00** |
| 47 | 0.00 | 0.00 | 0.00 | **1.00** | 0.00 | 0.00 | 0.00 | **1.00** |
| 50 | 0.00 | 0.00 | 0.00 | **1.00** | 0.00 | 0.00 | 0.00 | **1.00** |
| 51 | 0.00 | 0.00 | 0.00 | **1.00** | 0.00 | 0.00 | 0.00 | **1.00** |
| 55 | 0.00 | 0.00 | 0.00 | **1.00** | 0.00 | 0.00 | 0.00 | **1.00** |
| 60 | 0.15 | **0.80** | 0.00 | 0.05 | 0.00 | **0.95** | 0.00 | 0.05 |
| 73 | 0.18 | **0.61** | 0.00 | 0.21 | 0.00 | **0.91** | 0.00 | 0.09 |

| Node | Character 3 ARD model | | | Character 3 ED model | | |
|------|-----|--------|---------|-----|--------|---------|
| | – | Absent | Present | – | Absent | Present |
| 2 | 0.00 | 0.00 | **1.00** | 0.00 | 0.36 | **0.64** |
| 4 | 0.00 | **0.99** | 0.01 | 0.00 | **0.99** | 0.01 |
| 7 | 0.00 | **0.99** | 0.01 | 0.09 | **0.91** | 0.00 |
| 17 | 0.00 | 0.00 | **1.00** | 0.00 | 0.21 | **0.79** |
| 18 | 0.00 | 0.00 | **1.00** | 0.00 | 0.21 | **0.79** |
| 19 | 0.00 | 0.00 | **1.00** | 0.00 | 0.18 | **0.82** |
| 20 | 0.00 | 0.00 | **1.00** | 0.00 | 0.24 | **0.76** |
| 25 | 0.00 | 0.00 | **1.00** | 0.00 | 0.02 | **0.98** |
| 26 | 0.00 | 0.00 | **1.00** | 0.00 | 0.02 | **0.98** |
| 27 | 0.00 | 0.00 | **1.00** | 0.00 | 0.09 | **0.91** |
| 29 | 0.00 | 0.27 | **0.73** | 0.00 | 0.35 | **0.65** |
| 30 | 0.00 | **0.81** | 0.19 | 0.00 | **0.92** | 0.08 |
| 33 | 0.00 | 0.00 | **1.00** | 0.03 | 0.02 | **0.95** |
| 34 | 0.10 | 0.02 | **0.88** | 0.13 | 0.02 | **0.85** |
| 35 | **0.93** | 0.01 | 0.06 | **0.89** | 0.01 | 0.10 |
| 38 | **0.93** | 0.01 | 0.06 | **0.89** | 0.01 | 0.10 |
| 39 | 0.00 | 0.00 | **1.00** | 0.00 | 0.00 | **1.00** |
| 42 | 0.00 | 0.00 | **1.00** | 0.00 | 0.00 | **1.00** |
| 43 | 0.00 | 0.00 | **1.00** | 0.00 | 0.00 | **1.00** |
| 47 | 0.00 | 0.00 | **1.00** | 0.00 | 0.03 | **0.97** |
| 50 | 0.00 | 0.00 | **1.00** | 0.00 | 0.00 | **1.00** |
| 51 | 0.00 | 0.00 | **1.00** | 0.00 | 0.00 | **1.00** |
| 55 | 0.00 | 0.00 | **1.00** | 0.00 | 0.00 | **1.00** |
| 60 | 0.00 | 0.00 | **1.00** | 0.00 | 0.00 | **1.00** |
| 73 | 0.00 | 0.00 | **1.00** | 0.00 | 0.00 | **1.00** |

**Note:**
Rounded values of the posterior probabilities; the higher values in bold; (–) inapplicable characters. See correspondence between the node and the clades in the "Results" section.

showed an ambiguous state result for the ancestral nodes of the clades (Xantusiidae (Gerrhosauridae + Cordylidae)) [19], and Pleurodonta [60].

There was no conflict between the parsimony method and both models of the Bayesian approach (Table 5; Fig. S3; Table S3) used to reconstruct the ancestral state of Squamata [2] since the Bayesian analyses show a greater certainty for the simple extracolumella as the ancestral state (Table 5) although also show a minimum probability for the complex state. The ARD model reconstruction mostly agrees with the parsimony results except for the following exceptions. At the nodes for Gekkota [4], Pleurodonta [60], Pygopodidae [7], and Phrynosomatidae [73], the higher probability for the ancestral state is for the complex extracolumella, and for the first three clades (Gekkota [4], Pleurodonta [60], and Pygopodidae [7]) the lower probability is for the absence of it. The ancestral node of Phrynosomatidae [73] shows lower and similar probabilities for the simple columella and its absence. The ancestral node for the family Lacertidae shows a higher probability for the simple extracolumella and a lower probability for the complex one. At the ancestral nodes of (Amphisbaenidae + Trogonophidae) [38], and (Bipedidae ((Cadeidae + Blanidae) (Amphisbaenidae + Trogonophidae))) [35] there is great certainty for the elongated extracolumella state, as the probabilities are very low values for other states. The clade (Xantusiidae (Gerrhosauridae + Cordylidae)) [19] shows a high probability for the simple state and a lower probability for the complex state.

The ER model reconstruction mostly agrees with the parsimony results but shows the following differences (Table 5). In the ancestral node for Phrynosomatidae [73] there is a high probability for the complex columella state and a lower one for a simple columella; the ancestral node of (Amphisbaenidae + Trogonophidae) [38] has a major probability for the elongated state compared to a lower likelihood for the absent condition, but at the node for (Bipedidae ((Cadeidae + Blanidae)(Amphisbaenidae + Trogonophidae))) [35] the higher probability is the absence of extracolumella with lower values for the elongated and simple state. For the ancestral node of Pleurodonta, there is a greater certainty for the complex extracolumella; and for (Xantusiidae (Gerrhosauridae + Cordylidae)) [19] the higher value is for the simple state and the lower for the complex one. With the reconstruction of the ARD model, the ancestral node estimate for the family Lacertidae shows a higher probability for the simple extracolumella and a lower probability for the complex one.

### Character 3. Nature of the Internal Process

The parsimony reconstructions (Fig. 8B; Fig. S1) estimated the ancestral condition for Squamata [2] is the absence of internal process, which was also the reconstructed state for Gekkota [4] and Gymnophthalmidae [30]; while the evolutionary novelty, the presence of the process, was reconstructed in the ancestral nodes for Teiioidea [27], Lacertoidea [26], Anguimorpha [43], (Anguimorpha + Iguania) [42], Iguania [50], (Lacertoidea (Serpentes (Anguimorpha + Iguania))) [25], Anguidae [47], Acrodonta [51], Pleurodonta [60], and Phrynosomatidae [73]. This reconstruction shows as ambiguous states the ancestral nodes of the clades Scincoidea [18], (Xantusiidae (Gerrhosauridae + Cordylidae)) [19], Xantusiidae [20], (Scincoidea (Lacertoidea (Serpentes (Anguimorpha + Iguania))))

[17], and (Alopoglossidae + Gymnophthalmidae) [29]. The character is not applicable for amphisbaenians.

Contrary to the parsimony results, the reconstructions obtained for this character using the ARD (Table 5; Fig. S4; Table S3) model defined the presence of the internal process as the ancestral state of Squamata [2] with great certainty, while for the ER model (Table 5; Fig. S4; Table S3) it remains ambiguous, showing similar probabilities for both states (Table 5). The ARD model reconstruction mostly agrees with the parsimony results but shows the following exceptions. The presence of an internal process has a high probability in the reconstruction of the nodes of Scincoidea [18], (Xantusiidae (Gerrhosauridae + Cordylidae)) [19], Xantusiidae [20]; (Scincoidea (Lacertoidea (Serpentes (Anguimorpha + Iguania)))) [17]. This reconstruction results in ambiguous state estimations for the ancestral node of Gymnophthalmidae [30] with a higher probability for the absence than the presence of the internal process, while in (Alopoglossidae + Gymnophthalmidae) [29], the higher probability is for the presence. In the amphisbaenian clade [34], the highest likelihood is for the presence of the process and a lower probability for the inapplicability of the character, while the clades (Amphisbaenidae + Trogonophidae) [38], and (Bipedidae (Cadeidae + Blanidae) (Amphisbaenidae + Trogonophidae)) [35] show the contrary.

There are a few differences between the reconstructions obtained with the ER model (Table 5; Fig. S4) and the parsimony analysis (Fig. 8B; Fig. S1). The ER model found a higher probability for the presence of the process in the ancestral node of the clades Teiioidea [27] and Gymnophthalmidae [30]. For the nodes of the clades where the character is not applicable, the ER model found a higher probability for the presence of the process in the ancestor of amphisbaenians [34], contrary to the values found for the ancestral node of (Amphisbaenidae + Trogonophidae) [38] and (Bipedidae ((Cadeidae + Blanidae) (Amphisbaenidae + Trogonophidae))) [35]. The ancestral nodes of the clades (Amphisbaenidae + Lacertidae) [33], Lacertoidea [26], and Pygopodidae [7] show lower probabilities for the inapplicability of the character, with a higher probability for the presence of the process in the two first clades and the absence in the last one. The ER model analysis found a higher probability for the presence of the process in the ancestral nodes of the clades Xantusiidae [20], (Xantusiidae (Gerrhosauridae + Cordylidae)) [19], Scincoidea [18], (Alopoglossidae + Gymnophthalmidae) [29], (Scincoidea (Lacertoidea (Serpentes (Anguimorpha + Iguania)))) [17], that were defined as ambiguous by the parsimony approach.

## DISCUSSION

Although there is a lot of information available about the skull of lizards, most of these publications provide incomplete information about the middle ear, being limited to only a few details of the columella and even less about the extracolumella. The main studies regarding the middle ear as an anatomical complex, were realized by *Versluys (1898)* and *Wever (1973, 1978)*. These authors described morphological details of each structure for many species within a comparative framework that has allowed the establishment of morphological patterns of the middle ear of lizards. This study adds detailed information
about the middle ear morphology and variation in lizard, revealing an important source of variation previously understudied.

In general, lizards have a middle ear formed by a columella, and an extracolumella (which shows an internal process in some groups), with the latter structure displaying large morphological variation (*Wever, 1978*). Some species show extreme modifications or reductions of the middle ear (e.g., *Blanus* and *Bipes*, *Wever & Gans, 1973*; *Wever, 1978*; *Chamaeleo*, *Wever, 1968*; and *Rhampholeon*, *Toerien, 1963*), or even the total absence of it (e.g., *Aprasia* spp., *Baird, 1970*; *Wever, 1978*; *Daza & Bauer, 2015*).

## Columella

The typical pattern of the middle ear in lizards shows a quite conservative columella (*Wever, 1978*). However, in some cases, it is complicated to compare the scarce variation that it presents, due to the terminology used to describe this structure in the published descriptions.

The presence of the stapedial foramen (Fig. 2A) is accepted as a ancestral condition in reptiles (*Goodrich, 1958*; *Underwood, 1957*; *Greer, 1976*; *Estes, de Queiroz & Gauthier, 1988*; *Gauthier, Estes & de Queiroz, 1988*). The only living lepidosaurs that exhibit this foramen are *Anelytropsis*, *Dibamus*, and some gekkotans (*Kamal, 1961*; *Kluge, 1967*; *Greer, 1976*; *Rieppel, 1984*; *Estes, de Queiroz & Gauthier, 1988*; *Gauthier, Estes & de Queiroz, 1988*; *Bauer, 1990*). Although this foramen may be present in embryos of amphisbaenians, it is always absent in the adults (*Versluys, 1898*; *Gans, 1978*; *Kearney, 2003*). In gekkotans the foramen has been recorded in all genera of Sphaerodactylidae (*Bauer et al., 2018*), and some representatives of Eublepharidae (*Posner & Chiasson, 1966*), Gekkonidae (*Kluge & Eckhardt, 1969*; *Bauer, 1990*; *Daza, Aurich & Bauer, 2012*; *Villa et al., 2018*), and Phyllodactylidae (*Daza et al., 2017*; *Villa et al., 2018*). As expected, we recorded the presence of the stapedial foramen in all the gekkotans examined (Table 2), confirming its presence in *Gonatodes* (Sphaerodactylidae), *Hemidactylus* and *Phelsuma* (Gekkonidae), and *Tarentola mauritanica* (Phyllodactylidae), as previously registered by *Villa et al. (2018)* in this last species. We also confirmed the absence of the stapedial foramen in *Lialis* (Pygopodidae) and *Thecadactylus* (Phyllodactylidae), as was previously recorded by *Kluge & Nussbaum (1995)* and *Wever (1974)* for these genera. The absence of the stapedial foramen has also been recorded in several genera of Gekkonidae, such as *Christinus* (*Bauer, Good & Branch, 1997*), *Ebenavia*, *Gehyra*, *Gekko*, and *Paroedura* (*Kluge & Nussbaum, 1995*); and both states have been described in the genus *Homonota* (Phyllodactylidae) – absence by *Kluge & Nussbaum (1995)*, and presence by *Daza et al. (2017)*.

There are some relative differences in the size of the rod and footplate of the columella in lizards. According to *Wever (1978)*, the rod is usually slender and flexible, although in a few species it is thick and sturdy; and the footplate is mostly broadly flared, while a rounded knob footplate, a little larger than the rod itself, is present in just a few instances (*Wever, 1978*). *Evans (2008)* describes the sizes of the rod and footplate and its variation using the more common morphological pattern (referred to as the "normal" pattern) as a point of comparison: a slender rod with a small footplate, typical pattern exhibit by

iguanians. Thus, according to *Evans (2008)*, the columellar rod is: "normal" in iguanians, gekkotans, and scincids; shorter and usually with an expanded footplate, as in *Anguis*, *Saurodactylus*, *Xenosaurus* (*Rieppel, 1980*, Fig. 21), Agamidae, and Dibamidae; or longer, as in *Shinisaurus*. It can also vary from long to short within the same genus, as in *Ceratophora* (*Pethiyagoda & Manamendra-Arachchi, 1998*), or show tendencies towards the reduction of the rod and enlargement of the footplate, as observed in gymnophthalmids (*Evans, 2008*). In some of the previously published morphological descriptions, there are a few specific remarks made regarding the size of the columellar rod, such as noting the extremely short length in amphisbaenians (*Wever & Gans, 1973*), and the agamid *Ceratophora* (*Pethiyagoda & Manamendra-Arachchi, 1998*). Substantial differences in the increased size of the footplate have been frequently described, for example: the expanded stapedial footplate of amphisbaenians and anniellids (*Baird, 1970*; *Wever & Gans, 1973*), the noticeable asymmetrical footplate of *Draco volans* (*Wever, 1978*), and the large footplates of *Anniella pulchra*, *Cophosaurus texanus* (*Wever, 1973*), *Ceratophora stoddartii* (*Wever, 1978*), and *Rhineura floridana* (*Baird, 1970*; *Olson, 1966*). Most of the specimens examined in this study exhibit a slender columellar rod with a proportionally small footplate, except in the case of *Lialis jicari* (Fig. 3B) which shows an evident short, but not stout, rod with a small footplate. This description differs from that of *L. burtonis* by *Wever (1974)*, who described a short and sturdy columella with a relatively large footplate. In this case, according to the figure of the middle ear of *L. burtonis* (*Wever, 1974*, Fig. 4), it is possible to assume that there are no significant differences between the columella of *L. jicari* and *L. burtonis*, except in the references used to describe their sizes. It is difficult to compare the morphology of the columella between species due to the different parameters and criteria used by each author to estimate the size of the structures. For this reason, we chose to define a ratio between the size of the columella and one of its associated structures. Thus, given the functional role of the complex formed by the columella and extracolumella pointed out by *Wever (1978)*, we used the ratio between the relative length of the columellar rod and the length of the central axis of the extracolumella (Figs. 1, 2C, 3A and 3B), previously defined as ANC—"total anchorage length" by *Werner & Igić (2002)*. Using our observations and some illustrations available in the literature (see Table 4), we were able to estimate the different conditions of this feature in some species. We are aware that gathering information on this feature without precise measures, as well as estimating the measures from published illustrations is not the most accurate method; however, this provides some assessment regarding the existing variation in this ratio and affords a preliminary estimation of the evolutionary history of variation in this feature. Based on the current information available, there is no phylogenetic signal to the variation of the columella-extracolumella ratio we observed in the major groups of lizards, since the parsimony-based ancestral state reconstruction shows multiple independent appearances of all three states of this character in less inclusive groups, and the Bayesian approach found similar probabilities for each state at all ancestral nodes (Fig. 7; Table 5).

The expanded distal end of the osseous columella (Fig. 3) is not explicitly mentioned in the available descriptions of the lizard columella; however, *Wever (1978)* described and

illustrated a thin, delicate, and rather flexible mid-portion in the columella of *Trachylepis brevicollis* (= *Mabuya brevicollis*) that was also illustrated in other species, such as *Crotaphytus collaris, Callisaurus draconoides, Holbrookia maculata*, and *Sceloporus magister* (*Wever, 1978*). These records make evident the observation of a widening of the distal end of the columella in these species, a feature that we also registered in some species (see Table 2). *Werner & Igić (2002)* measured different elements of the middle ear to establish the effects of the dimensions of these structures on the auditory sensitivity of gekkonid lizards. Their results suggest that part of the sensitivity in these lizards would depend on the sizes of the structures of the middle ear. The columella measures used in that study were: the length of the columella and its diameter in the midpoint, and the diameter of the footplate (*Werner & Igić, 2002*, Fig. 1). Thus, the presence (Fig. 3) or absence of a widening in the distal end of the columella could also be related to auditory sensitivity. However, our observations show the existence of both states of this feature (presence and absence of the widening) in *Anadia bogotensis*, implying this trait displays individual variation, and hence we flag the necessity of evaluating this feature across a larger sample of individuals.

According to *Wever (1978)*, in some species, the cartilaginous joint between columella and extracolumella shows a discontinuity comprised of dense connective tissue that gives rigidity to this point, and that can surround the joint, or occur between both structures. Apparently however, the only specific record of this feature was made by *Wever (1978)* mentioning the absence of this kind of joint in *Trachylepis brevicollis* (= *Mabuya brevicollis*). In our study, both the presence and absence of the connective tissue in this joint were observed in different groups and families (Table 2), and even in the same species, *Anolis marianum*, which suggests this feature possibly displays intraspecific variation. With the current data, we cannot address the amount of variation, thus it is necessary to examine more specimens of *Anolis marianum* to establish if it could be due to ontogenetic variation or a polymorphism that could support the presence of cryptic species. We also suggest making an in-depth exam using more detailed sampling methods, such as histological techniques, to confirm the kind of tissue involved and determine its definite association with both the columella and the extracolumella.

## Extracolumella

Several descriptions and illustrations of the extracolumella exist, which present accurate and detailed information and show significant morphological variation of this structure (e.g., *Versluys, 1898*; *Peterson, 1966*; *Posner & Chiasson, 1966*; *Wever, 1968*; *Wever & Werner, 1970*; *Werner & Wever, 1972*; *Wever, 1973*, *1978*; *Werner et al., 2005*). Some variations of the extracolumella are relatively rare, such as the extreme reduction observed in *Varanus bengalensis* (Varanidae, *McDowell, 1967*); a distinct rough oval form in *Lanthanotus borneensis* (Lanthanotidae, *McDowell, 1967*); a short structure with a dense mass of ligamentous fibers that split into two branches, one extending along the lower jaw, and the other along the upper jaw in *Rhineura floridana* (Rhineuridae, *Wever, 1978*); and an elongated structure that extends along the quadrate and laterally connects with the labial skin in Amphisbaenidae and Trogophidae, (*Versluys, 1898*; *Wever & Gans, 1973*;

*Kearney, 2003*; *Kearney, Maisano & Rowe, 2005*). The absence of the extracolumella in lizards has only been registered in the species of *Aprasia* (Pygopodidae, *Wever, 1978*), *Bipes* (Bipedidae, *Wever & Gans, 1973*), and *Blanus* (Blanidae, *Wever & Gans, 1973*). On the other hand, the more common morphological pattern found in lizards is an extracolumella with four principal processes. Some of the variation described for this element refers to the size or lack of one or more of these processes. In most species, all these processes are easily distinguished, but in a few cases, as in *Ceratophora stoddartii* (Agamidae) and *Chamaeleo* (Chamaeleonidae), there is some uncertainty about a processes' presence and equivalences (*Wever, 1973*, *1978*).

The four extracolumellar processes have been either described or illustrated in *Callisaurus* (Phrynosomatidae); *Coleonyx variegatus* and *Eublepharis macularius* (Eublepharidae), *Chondrodactylus bibronii* (= *Pachydactylus bibronii*) and *Gekko gecko* (= *Gekko verticillatus*) (Gekkonidae); *Crotaphytus collaris* (Crotaphytidae); *Iguana iguana* (= *Iguana tuberculata*) (Iguanidae); and *Lialis burtonis* (Pygopodidae) (*Versluys, 1898*; *Iordansky, 1968*; *Posner & Chiasson, 1966*; *Werner & Wever, 1972*; *Wever, 1974*, *1978*; *Werner et al., 2005*). In this study, we found these four processes to be present in Agamidae, Dactyloidae, Hoplocercidae, Lacertidae, Phyllodactylidae, Sphaerodactylidae, and Tropiduridae, and in two additional species of Gekkonidae and one of Pygopodidae (Table 3). In all these cases, the pars superior and inferior, and the anterior and posterior processes are evident and easily recognized. The presence of the four processes registered here in the species of Gekkota agrees with the literature records for this group, and we also add information on these features to the morphology previously described in Agamidae and Lacertidae (see below).

The absence or extreme reduction of the pars superior only has been registered in *Draco volans* and *Phrynocephalus maculatus* (Agamidae), and *Cophosaurus texanus* (Phrynosomatidae) (*Wever, 1973*, *1978*), and there are no records indicating the absence of the pars inferior in any of the lizard groups. In contrast, the lack of the anterior, posterior, or both processes are more frequent within some families and genera. In Gymnophthalmidae, the genera—*Anadia*, *Gelanesaurus*, *Neusticurus*, *Riama*, and *Tretioscincus* do not have an anterior process; while *Loxopholis* lacks both processes (Table 3). In Teiidae, the genera—*Pholidoscelis lineolatus* (= *Ameiva lineolata*), and *Tupinambis teguixin* (= *T. nigropunctatus*) do not have the anterior process (*Versluys, 1898*; *Wever, 1978*), while *Cnemidophorus lemniscatus* lacks both processes. In Lacertidae, there is no anterior process present in *Timon lepidus* (= *Lacerta ocellata*) (*Versluys, 1898*), but we recorded the presence of a very short and thin anterior process in *Acanthodactylus* cf. *schmidti*. The agamids *Draco volans* and *Phrynocephalus maculatus* do not have any of these processes (*Wever, 1973*, *1978*), and this feature corresponds to our observations in *Stellagama stellio*, but differs from those in *Acanthocercus atricollis* and *Leiolepis belliana*, species that exhibit all four extracolumellar processes. The variation in this structure has also been described within some genera. According to *Earle (1961a*, *1961b*), the genera *Callisaurus* and *Holbrookia* (Phrynosomatidae) have four extracolumellar processes, while *Wever (1973*, *1978*) points out that *C. draconoides* and *H. maculata* do not have either the anterior nor the posterior processes. Furthermore,

*H. maculata* also shows an extreme reduction of the pars superior and inferior. Similarly, according to *Wever (1973)*, and *Han & Young (2016)*, *Phrynosoma coronatum* (Phrynosomatidae) and *Varanus salvator* (Varanidae) do not present the anterior process; while *Versluys (1898)*, *McDowell (1967)*, and *Wever (1973)* stated that *P. platyrhinos*, *V. bengalensis*, and *V. niloticus* do not exhibit either process. We observed interspecific variation in *Pholidobolus* (Gymnophthalmidae), since *P. montium* does not have the anterior process and *P. vertebralis* does not have either of them.

The absence of both processes, anterior and posterior, has been recorded in *Anguis fragilis* and *Anniella pulchra* (Anguidae), and *Trachylepis brevicollis* (= *Mabuya brevicollis*) (Scincidae) (*Versluys, 1898*; *Wever, 1973*, *1978*). We found this condition in *Cnemidophorus lemniscatus* (Teiidae) and the species of *Mabuya* (Scincidae). The absence of the posterior process, when the anterior process is present, has only been reported in *Heloderma suspectum* (Helodermatidae) and *Xenosaurus grandis* (Xenosauridae) (*Versluys, 1898*; *Wever, 1973*, *1978*).

The available information about the shapes of the extracolumellar processes describes them as pointed and long or short cartilaginous structures, without any further descriptive detail. There are no specific descriptions of the shape of each extracolumellar process, except for a few mentions and illustrations of the anterior process in some species of Gekkota (*Versluys, 1898*; *Posner & Chiasson, 1966*; *Werner & Wever, 1972*; *Wever, 1978*; *Werner et al., 2005*, *2008*). In the specimens available for this study, we found some differences in the shapes of the extracolumellar processes, which illustrates wide variation in these structures. Although our sample is not representative of all groups of lizards, it was enough to display such variation, mainly in the pars superior and the anterior process. Thus, with the available information, the pars superior, which shows noticeable variation in its shape (Table 3), characterizes the species of Gekkota with a posterior prolongation of its upper edge (Figs. 4A–4C, 5A); while Hoplocercidae (Figs. 5B and 6B) can be differentiated by a rounded upper edge; Scincidae (Fig. 6C) by a tridentate upper edge; and *Tropidurus pinima* (Tropiduridae) by an anteriorly prolonged and shorter upper edge (Fig. 5C).

Among the species studied which show an anterior process, the more frequently observed shape is a pointed cartilaginous extension that can be short (Fig. 3C), or long (Figs. 4C, 5B and 5C), which corresponds with the shape most commonly described in the literature. However, we found that in the specimens of Gekkonidae and Phyllodactylidae examined (Table 3), the anterior process is a long and thick extension with some small and sharp prolongations (Figs. 4A and 4B). This shape has also been described or illustrated in Eublepharidae (*Coleonyx variegatus, Eublepharis macularius*), and Gekkonidae (*Chondrodactylus bibronii* and *Gekko gecko*) (*Versluys, 1898*; *Posner & Chiasson, 1966*; *Werner & Wever, 1972*; *Wever, 1978*; *Werner et al., 2005*). The remaining species of Gekkota examined (Table 3) did not show these sharp prolongations in the anterior process. One example is *Lialis jicari* (Pygopodidae, Fig. 4C), which shows a long and pointed process that is not oriented anteriorly, but downward; as well as the distal end of the anterior process that turns downward in *Gonatodes* (Sphaerodactylidae, Fig. 5A).

The pars inferior and the posterior process are more morphologically conserved. The pars inferior shows a sharp distal end in most of the species with available information, but a thicker distal end in Gekkonidae, Phyllodactylidae, and Sphaerodactylidae (Table 3). In the posterior process, the only variation observed was the overall size, except in *Lialis jicari* that shows both a short and thick posterior process that turns upward resembling a hook (Fig. 4C). These features—the shapes of the pars superior, the anterior process, and the shape of the distal end of the pars inferior— should be evaluated in greater detail and in a larger sample, to confirm if the variation observed has any taxonomic relevance within Gekkota.

## The internal process

The internal process is an additional extracolumellar structure that arises close to the joint with the columella, running anteriorly to attach to the quadrate. The proposed function of this process is mainly to protect the middle ear structures (*Wever, 1978*). The internal process was very similar in all species studied. It is fan-shaped, and the main morphological variation was the width of its origin at the shaft of the extracolumella. The shape of the process is similar to the morphology described by *Wever (1978)* in *Sceloporus magister* (Phrynosomatidae*), Crotaphytus collaris* (Crotaphytidae), *Ameiva lineolata* (= *Pholidoscelis lineolatus*, Teiidae), and *Agama agama* (Agamidae), but it is not possible to compare the extracolumellar origin of the process based on the Wever's descriptions. *Wever (1978)* differentiated two internal process types based on an auditory experiment's results and the process's flexibility and shape. The experiments consisted of measuring the columella sensitivity to a range of tones with two different variations, the internal process attached to the quadrate (its normal condition) and with this connection interrupted. Results on the experiments of *C. collaris* were similar, showing a slight improvement in the responses to low tones and a slight decrease to high ones. In *C. collaris*, it appears that the role of the internal process is related to protection rather than hearing. However, in other species such as the phrynosomatid *Callisaurus draconoides* (*Wever, 1978*: Figs. 6–19 and 6–20) where the internal process is less flexible or it "consists of a substantial mound-like elevation" (*Wever, 1978*: 158), the results of the experiment showed some differences when the connection of the internal process with the quadrate was interrupted. The sensitivity did not show major changes to low frequencies but showed a significant effect in losing the sensitivity to high frequencies, suggesting that the internal process has an auditive function (*Wever, 1978*). According to this, the morphology and the function of the internal process must be evaluated in more detail. Given the great diversity of the groups that have an internal process, it is expected that there will be significant variation among the groups.

## The middle ear types in lizards

The three types of middle ear described by *Wever & Werner (1970)* represent the more common morphologies observed in lizards and show an important morphological variation within each one. Despite the morphological differences between the types, all of these are highly effective in sound reception and transmission (*Wever, 1973*). According to

*Wever (1978)*, the most common type in lizards is the iguanid type that is present in Iguanidae, Agamidae, Cordylidae, Gerrhosauridae, Helodermatidae, Lacertidae, Teiidae, Varanidae, and Xantusiidae (see *Wever, 1978*, Table 5-III, p. 132). The species that *Wever (1978)* originally included in Iguanidae now belong to the families Corytophanidae, Crotaphytidae, Dactyloidae, Tropiduridae, Opluridae, Phrynosomatidae, and Iguanidae (see *Wever, 1978*, pp. 215–216). In addition, in our work, we found this pattern in species from some of these families and from Hoplocercidae (Table 3) that we add to the list. According to *Wever & Werner (1970)*, the iguanid type is characterized by the presence of the internal process. To this, we add that this type is further characterized by the presence of at least three well-defined extracolumellar processes, since all species that exhibit the internal process also have these additional processes. Given the variation observed in the shape and number of the extracolumellar processes within the iguanid type, we suggest greater evaluation of these characters within the families that possess them, in order to determine whether the variation in the morphology of these processes provides further systematic information at a finer taxonomic scale.

The gekkonid middle ear type is only present in the families of Gekkota (*Werner & Wever, 1972*; *Wever, 1978*). Although we did not have available material to check the presence of the extracolumellar muscle in any specimen within our sample, we recorded that none of the species of Gekkota studied showed internal processes. Additionally, all the specimens from these families exhibited: (i) four extracolumellar processes, (ii) a posterior extension in the pars superior, and (iii) an anterior process with some small and sharp projections. Thus, we add these three features to the definition of the gekkonid type described by *Wever & Werner (1970)*. The posterior extension of the pars superior and the shape of the anterior process and its projections could be diagnostic characters for Gekkota, and the variation present within these features may even be further diagnostic within the group as well. For this reason, we recommend more detailed analysis in a systematic context.

The simplest type of the middle ear is that of the scincids, which was described in Scincidae, Anguidae, and Xantusiidae (see *Wever, 1978*; Table 5-III). Interestingly however, the family Xantusiidae actually shows two different middle ear types: the scincid type is seen in *Lepidophyma flavimaculatum* and *L. smithi*, that do not possess both the internal process and the extracolumellar muscle; and the iguanid type is observed in *Xantusia henshawi*, which does have the internal process (*Wever, 1978*). The absence of the extracolumellar muscle was not evaluated in the latter species, but the absence of the internal process was corroborated here in the genus *Mabuya* (Scincidae).

The "divergent" or "degenerate" (as called by *Wever (1978)*) middle ears are those with a morphology that does not match with any of the three previously mentioned types (*Wever & Werner, 1970*; *Wever, 1973*, *1978*). However, all genera described by *Wever (1973)* as divergent forms, except those in the genus *Anguis*, exhibit an internal process, which is small and, in some cases, extremely reduced (*Wever, 1973*). According to *Wever (1978)*, divergent middle ears are present in Chamaeleonidae, and Xenosauridae, as well as in some species of Agamidae and Scincidae, and less frequently in some species of the families Anguidae, Pygopodidae, Teiidae, and in several families of Iguania

(*Wever, 1978*; Table 5-III). The genus *Feylinia* and the families Dibamidae and Lanthanotidae also show this type of middle ear (*McDowell, 1967*; *Baird, 1970*; *Wever, 1978*). The genera *Anguis*, *Anniella*, *Callisaurus*, *Ceratophora*, *Cophosaurus*, *Draco*, *Holbrookia*, *Phrynocephalus*, *Phrynosoma*, and *Xenosaurus* show a divergent pattern (*Wever, 1973*). All of them lack the tympanic membrane and exhibit an extreme reduction in the extracolumella.

## Ancestral state reconstructions

Ancestral state reconstructions of the available information indicated that at least some extracolumella features can be a useful source of systematic information within Squamata. The great uncertainty shown by the analyses for the ancestral state of the length of the columella relative to the extracolumella central axis length (character 1, Fig. 7) suggests that there is no phylogenetic signal associated with this feature. The parsimony analysis shows an ambiguous ancestral node between the longer and shorter states, while there are no differences between the results of Bayesian models ARD and ER where the probability of the ancestral condition is equal for all states (Table 5; Figs. S1 and S2). The variation observed in this ratio could be related to the auditory sensitivity associated with the inner ear, as well as morphological or morphometrical features of the skull and the outer ear, or even ecological conditions.

To understand the evolutionary history of the extracolumella, the different morphological variations and the particular shapes of its processes should be evaluated in more detail and within less inclusive groups. However, simplifying the available information into only four states: extracolumella simple, complex, elongated, and absent (character 2, Fig. 8A) provides at least a broad idea of the overall variation and the general evolutionary history of the extracolumella in lizards. While the presence of a simple extracolumella is the ancestral condition of Squamata, the complex extracolumella appears to have arisen via convergence in Gekkota, Pleurodonta, and Xantusiidae, and could be a diagnostic character (along with other features) for members of these groups. The families Agamidae, Lacertidae, and Phrynosomatidae are polymorphic in that different members of these clades exhibit a simple or complex extracolumella (Fig. 8A). Although there are four extracolumellar processes exhibited in Xantusiidae (*Wever, 1978*), Agamidae, and Lacertidae (this study), the anterior process in the first family and the anterior and posterior processes in the latter two, are extremely small and thin structures, giving a similar appearance to the simple extracolumella, emphasizing the necessity for detailed observation in species that apparently lack any processes.

The elongated extracolumella is extremely different morphologically and is present only in Amphisbaenidae and Trogonophidae. It is a cartilaginous structure that runs anteriorly along the quadrate and is attached to the skin which functions as a sound-receptive surface (*Wever & Gans, 1973*; *Wever, 1978*). The origin of the amphisbaenian extracolumella has been a controversial topic since *Fürbringer (1919*, *1922)* proposed that it originated from the epihyal portion of the hyoid apparatus, while *Camp (1923)* stated that these structures are not related. Later, based on their personal observations, *Wever & Gans (1972*, *1973)* supported Fürbringer's proposal, suggesting that

the amphisbaenian extracolumella is not homologous with that of lizards, but instead is a modification of a dorsal portion of the hyoid (see *Wever & Gans, 1973*). However, according to *Kearney (2003)*, this hypothesis has not been tested since there are no studies about the development of amphisbaenians that have found any relation between the extracolumella and the hyoid. Considering the statement of *Kearney (2003)*, we consider the extracolumella of Amphisbaenidae and Trogonophidae as a structure homologous with the lizard extracolumella. Whenever it is present, the extracolumella always connects with the dermal layer of the skin in members of the amphisbaenian clade. Aside from this however, members of this group exhibit wide variation in extracolumellar morphology. This variation is present in the family Rhineuridae that despite having a reduced extracolumella, also exhibits an unusual morphology in that it has two branches of ligament fibers—one connected with the lower jaw and the other with the upper jaw (*Wever, 1978*). Another kind of variation is present in *Diplometopon zarudnyi* (Trogonophidae) whose extracolumella has a triangular blade shape extending anteriorly over the skull's lateral surface with its posterior third cartilaginous and a heavily calcified outer surface (*Gans & Wever, 1975*). In these species, the sound-receiving surface is not a tympanic membrane but a particular cephalic scale area. Sounds are transmitted through the ground, and their vibrations are detected when the specimen has its head in contact with the substrate (*Wever & Gans, 1972*, *1973*). These modifications are part of a suite of advantageous features for a fossorial lifestyle in amphisbaenians (*Baird, 1970*; *Wever & Gans, 1972*, *1973*).

The genus *Aprasia*, and the families *Bipedidae* and *Blanidae*, do not have extracolumellas indicating at least two independent losses of the extracolumella in Squamata. The genus *Aprasia* does not have a tympanic membrane, a columellar apparatus, or a tympanic cavity (*Baird, 1970*; *Wever, 1978*), although some species might have a small tympanic membrane and a very rudimentary columella. The morphology of the inner ear and some anatomical modifications in the pterygoid and quadrate of *Aprasia repens* denote normal auditory function, where the quadrate plays a role in sound transmission (*Daza & Bauer, 2015*). These observations suggest a limited ability to hear airborne sounds, but also potential capacity to hear "underground sound" (*Greer, 1989*; *Daza & Bauer, 2015*). In *Aprasia repens* (Pygopodidae), the pterygoid and quadrate bones are the ones that show the morphological modification to favor the auditory function in this burrower gecko. Low-frequency vibrations are intercepted by the lower jaw, and its transmission into the middle ear might be through the quadrate. The pterygoid is not in contact with the quadrate to prevent the entrance of the vibrations into the palate (*Daza & Bauer, 2015*). The ear modifications are one distinctive feature of the extremely divergent morphological condition of the fossorial adaptation that this genus shows (*Baird, 1970*). The loss of the extracolumella also occurred in the ancestor of the clade (Bipedidae + (Blanidae + Cadeidae) (Amphisbaenidae + Trogonophidae)), but it appears again as an expanded structure in Amphisbaenidae and Trogonophidae. In this clade, we could expect that Cadeidae, a family with no current information, does not have an extracolumella (see below), similar to *Bipes* (Bipedidae) and *Blanus* (Blanidae) that lack the external ear and only have a columella that ends in a disk of fibrous

tissue beneath the skin, resulting in a very aberrant sound receiving system, but with a high level of sensitivity stimulated by aerial sounds (*Wever & Gans, 1972*, *1973*).

In the ancestral reconstruction of character 2 (Extracolumella), the results of the ARD and ER Bayesian approaches show some differences in the probability values for the ancestral state estimates for the clades Gekkota, Pleurodonta, and Xantusiidae. However, both analyses show the highest support for the complex extracolumella at the ancestral node of the three clades, consistent with the parsimony results (Table 5; Fig. S3). A second difference between the two Bayesian analyses was in the probability values of the nodes within the amphisbaenian clade. In this case, both analyses still estimated the highest probability for the elongated extracolumella at the ancestral node of (Amphisbaenidae + Trogonophidae), agreeing with the parsimony results. Contrary to this, the ARD model shows the highest probability values for the elongated extracolumella in the ancestral nodes of (Bipedidae + (Blanidae + Cadeidae) (Amphisbaenidae + Trogonophidae)), ((Blanidae + Cadeidae) (Amphisbaenidae + Trogonophidae)), and (Blanidae + Cadeidae), suggesting the presence of an elongated extracolumella in Cadeidae. In contrast, the ER model, concordant with the parsimony results, shows the highest support for the absent extracolumella at the ancestral nodes for these clades, proposing the absence of an extracolumella in Cadeidae (Table 5; Figs. S1 and S3).

Serpentes have a long and narrow columella with a cartilaginous end that connects with the quadrate through an articulatory process, and in some groups, intermediate cartilages may also be observed between both structures (*Wever, 1978*). The identity of the cartilaginous columella end, as well as the intermediate cartilages, is uncertain. According to *Rieppel & Zaher (2000)*, the columella's cartilaginous end may be homologous to the internal process rather than the main body of the extracolumella. Furthermore, according to *Kamal & Hammouda (1965)*, the intermediate cartilages are intercalary structures between the articular process and the cartilaginous end of the columella, while *McDowell (1967)* considered these as the internal process of the columella and a piece of the extracolumella. Since there is no consensus about the nature of the extracolumella in Serpentes and that this subject is beyond the focus of this study, we cannot make any assumptions about this. Nevertheless, it is fundamental to define the cartilages' identity related to Serpentes' columella end and study its variation, to establish a more accurate hypothesis about the evolutionary history of the extracolumella in Squamata.

The ancestral reconstruction of the internal process (character 3; Fig. 8B) shows differences between analyses that do not permit establishing the ancestral state (presence or absence) for this character for Squamata, along with some of the other more ancestral nodes within this group (Fig. 8B). The absence of this process is likely a result of convergence occurring between the groups of Gekkota, Gymnophthalmidae, and Scincidae (Fig. 8B); while the presence of this process is the more common state within Squamata. Based on the available information, the families Anguidae and Xantusiidae are the only ones which are polymorphic for this character state. The result of the parsimony analysis indicated the absence of the internal process as the ancestral condition of Squamata, but the Bayesian analyses differs from it. The ARD model result shows the presence of the internal process as the ancestral condition, while the ER model show

similar probabilities between absence and presence of this process (Table 5; Figs. S1 and S4). For the Gekkota, Gymnophthalmidae, and Scincidae clades, the results of the different analyses of the ancestral reconstruction agree, showing as the ancestral condition with the absence of the internal process in these groups (Table 5; Figs. S1 and S4).

The fossil record shows that the middle ear of the ancestral lepidosaurs have a tympanic membrane and that the lack of this structure in *Sphenodon* is the result of a secondary loss, possibly related to feeding specializations (*Evans, 2016*). There are few details about the morphology of the middle ear of stem squamates. According to *Evans (2016)*, the squamate fossil record from the Early Cretaceous with well-preserved skulls only shows evidence of the ear anatomy by the presence of a quadrate with a lateral conch and tympanic crest. Nevertheless, one specimen of the Early Cretaceous lizard, *Liushusaurus acanthocaudata* (*Evans & Wang, 2010*), shows traces of the cartilaginous extracolumella that lie adjacent to the tympanic region (*Evans, 2016*). The fossil record shows the derived condition, indicating that squamates evolved the tympanic ear according to their different specialized lifestyles (*Evans, 2016*).

The columella and extracolumella morphologies have not been associated functionally with lizards' vocalizing capabilities. However, given the high morphological complexity of the extracolumella described in the geckos' clade, it could probably be correlated with the vocalizations that they produce which are complex and exhibit variation in amplitude and frequency (*Russell & Bauer, 2020*). On the other hand, *Wever (1978)* considered a correlation between the vocalization and the meatal closure muscle of the outer ear in these lizards. According to *Wever (1978)*, the function of the meatal closure muscle is to protect the ear; although it is not clear if this protection is only against mechanical damage or also against particularly loud sounds. This muscle could be related to the fact that these lizards produce vocalizations, and hence the muscle plays a role in protecting the individual's ears against its own vocal sounds, which can be extremely loud in some species. However, in some individuals of the family Sphaerodactylidae and the gekkonid genus *Phelsuma*, which are considered to be mute species, or with tenuous vocalization, don't have this muscle; other species (e.g., *Gehyra variegata*, *Oedura monilis* (= *Oedura ocellata*), and *Strophurus elderi* (= *Diplodactylus elderi*)) that also do not produce vocalizations, do have the meatal closure muscle in their outer ears (*Wever, 1978*). Thus, while the production of loud vocalization might be related to the presence of the meatal closure muscle, it is clear that other conditions may also produce the development of this muscle (*Wever, 1978*). Alternatively, it can be assumed that the presence of the meatal closure muscle and vocalization are the ancestral condition for gekkotans, and in some groups the muscles have been lost along with vocalization, whilst in others the muscles haven't been lost yet. We cannot also rule out that this muscle has an unknown alternative function. The combined analysis of morphological and functional information is necessary to establish the possible relation between the outer and middle ear with geckos' vocalizations.

Despite the general morphology of the lizard middle ear being quite well known, and there being no particularly notable variation in the lizard columella, the morphological variation of the extracolumella structure is evidently more significant than previously

described. We have presented evidence of that extensive variation here and demonstrated that some features of the extracolumella could potentially provide a source of phylogenetic information for some groups. However, in some clades, other ear modifications may be more closely related to adaptations for navigating and functioning within particular habits. It is necessary to perform a more detailed and comprehensive study around each of the specific morphologies of the extracolumella, here defined as: simple, complex, and elongated, to understand better the variation present within each particular clade. This kind of detailed information will possibly let us know about more morphological features that may be useful to the systematic and understanding of the functioning of the middle ear in certain groups of lizards.

## CONCLUSIONS

The middle ear in lizards shows considerable morphological variation. Although the columella morphology is more conservative, the structures that conform to the extracolumella show a larger amount of variation than previously described, mainly in the pars superior and anterior process. Significant morphological variation of the internal process is expected given the vast diversity of the species that present this process and the evidence of a possible functional variation. These extracolumellar structures should be studied in more detail to complete as much as possible the gap of the information, especially within lizards' groups that have a complex extracolumella, which may present considerable morphological variation. Even though this study describes the variation of these structures only in some lizard species, this information gives us an idea about the amount of morphological variation that we could find across the Squamata. The analysis of this morphology within a comparative and evolutive framework shows us that these structures are a substantial source of systematic and phylogenetic information, which could be useful even to functional studies. The results of the ancestral reconstruction show high levels of homoplasy in the variation of the columella-extracolumella length ratio, while pointing out as the ancestral condition of Gekkota, Pleurodonta, and Xantusiidae the presence of a complex extracolumella; and in Gekkota, Gymnophthalmidae, and Scincidae, the absence of the internal processes. Furthermore, we can consider as diagnostic characteristics of Gekkota the presence of a posterior extension in the pars superior and an anterior process with some small and sharp projections. A more accurate description of each process of the extracolumella and its variation within less inclusive groups should be evaluated in more detail to establish the taxonomic and systematic value of these features. There is not enough information about the condition of the middle ear structures studied here to cover the complete clade of squamates, for that reason the only ancestral condition defined to this group was a presence of a extracolumella with less than four processes. The morphological variation of both the columella and extracolumella may have a distinctive role associated with their efficiency in transmitting the sound, and with the vocalizations produced by some clades. Also, the variation of the extracolumellar structures probably is correlated with different morphological patterns

of the outer ear, which at the same time are related to the specific habitats of each squamates group. These correlations should be established by studying the morphological and functional association between the middle and outer ear with the vocalizations within an ecological context.

## ACKNOWLEDGEMENTS

We would like to thank Martha Lucia Calderón (Instituto de Ciencias Naturales ICN, Universidad Nacional de Colombia), Juan Manuel Daza (Collection of Reptiles of the Museo de Herpetología MHUA, Universidad de Antioquia), and Hussam Zaher and Aline Staskowian Benetti (Museu de Zoologia MZUSP, Universidade de São Paulo), for providing access to specimens under their care. We also like to express our thanks to Rebecca Laver (Australian National University) and Stephanie Baker (Sam Houston State University) for their valuable comments to the previous version of this manuscript, and to Julia Schultz and Michael Caldwell for their insightful review of the work. PMSN is grateful to Dione Seripierri and the team of the Library of the MZUSP for your invaluable help in obtaining some of the key bibliographical references.

### Funding

This work was supported by the Pontificia Universidad Javeriana (No. ID 00007749 and PUJ-00200-18) and the Department of Biological Sciences at Sam Houston State University. The Office of Research and Sponsored Programs and the Department of Biological Sciences at Sam Houston State University covered the publications costs. The funders had no role in study design, data collection, and analysis, decision to publish, or preparation of the manuscript.

### Grant Disclosures

The following grant information was disclosed by the authors:
Pontificia Universidad Javeriana: ID 00007749 and PUJ-00200-18.
Department of Biological Sciences at Sam Houston State University.
Office of Research and Sponsored Programs.

### Competing Interests

The authors declare that they have no competing interests.

### Author Contributions

- Paola María Sánchez-Martínez conceived and designed the experiments, performed the experiments, analyzed the data, prepared figures and/or tables, authored or reviewed drafts of the paper, and approved the final draft.
- Juan D. Daza conceived and designed the experiments, analyzed the data, prepared figures and/or tables, authored or reviewed drafts of the paper, and approved the final draft.
- Julio Mario Hoyos conceived and designed the experiments, analyzed the data, authored or reviewed drafts of the paper, and approved the final draft.

## Data Availability

The files used in the analyses (topologies and matrices), figures, and numeric values from the results are available at MorphoBank Project 3551: Comparative anatomy of the middle ear in some lizard species with comments on the evolutionary changes within Squamata. Project DOI 10.7934/P3551.

## Supplemental Information

Supplemental information for this article can be found online at http://dx.doi.org/10.7717/peerj.11722#supplemental-information.

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
