# Peer review of "Comparative anatomy of the middle ear in some lizard species with comments on the evolutionary changes within Squamata"

_PeerJ, doi:10.7717/peerj.11722_

## Round 0.1 · original submission · Major Revisions

· Academic Editor

Major Revisions

Dear authors,

Thank you for your submission to PeerJ. I found this manuscript quite interesting and agree with reviewers that after revisions, it will make an excellent contribution to PeerJ.

Reviewer 1 notes that the title and some of the conclusions should reflect the relatively small sample size in the data. Per reviewer 2, I would also strongly encourage you to clarify your selection of character states and discuss them more throughly. Reviewer 2's comments relating to expanding the discussion by a paragraph to compare your ancestral state reconstructions to what is known in the fossil record would also be most welcome.

When you submit your revised manuscript, please include a clean version of the manuscript, a tracked changes version showing changes made, and a itemized reviewer response.

Please let me know if you have any questions. Thank you for your submission.

Best,

Brandon P. Hedrick, Ph.D.

·

Basic reporting

I have made numerous comments on the manuscript and will direct the authors and the editors to the marked pdf copy of the manuscript for details.

However, to summarize, in terms of "Basic Reporting", the manuscript suffers from two current flaws:

1) The authors empirical observations are limited to 38 species from four major groups of squamates (iguanians, scincids, geckos and lacertoids = ~4000 species of extant squamates). They did not sample snakes, anguimorphs, amphisbaenians, dibamids, xantusiids, etc., etc., but yet title their paper as "comparative anatomy" of "lizards". This is simply not the case as a sample of all lizards (remember, snakes are lizards too), let alone all lizard groups/clades, was not done. The study is even more limited than just 38 species as 9 of them are all from a single iguanian genus, Anolis, which significantly narrows again the broad view of all lizards/squamates. I hasten to point out that this is not a problem for the study, only for the manuscript, and it is easily remedied by rewriting the title to describe what the authors have actually done and outline those limitations when moving to their more broad conclusions. This is a positive change as it indicates that substantial work remains but that preliminary findings show promising results.

2) The authors highlight very nicely the anatomy of the middle ear apparatus of "some lizards" and create 3 characters for phylogenetic analysis and subsequent character mapping to understand middle ear evolution. Good. However, the manuscript focuses on only two of those three characters - the columella and extracolumella. The authors create a third character for the internal process, but devote only a handful of sentences to describing the morphology of literally 1/3 or 33% of their character complexes. The internal process requires a great deal more description and highlighting in the body of their manuscript. At the moment it is treated as an afterthought. For example, in the Discussion, there are secondary headings for the Columella and Extracolumella, and literally nothing for the internal process. In the Discussion, the internal process merits 6 sentences, yet as I said, it is 33% of the characters analyzed.

Experimental design

The design is fine, but the conclusions and title of the paper need to reflect the small size of the sample - 38 species from ~10,000, with 1/4 of the data coming from a single genus. I have no concerns with the methods as described and conducted. The investigation is rigorous, but limited in scope. There is still excellence in the work, but it needs to identify more precisely the limited scope.

Validity of the findings

All the data have been provided, they are robust, but need clarity in terms of limits relative to the scope of the project as written. All underlying data are provided.

Conclusions are more broad than the data allow, but this is easily remedied. See marked pdf and opening and closing comments in this review.

Additional comments

I am recommending major revisions, and am happy to review a revised draft of the manuscript. I do not think that the authors need to collect more data, nor change the way they have presented their results. However, I think the manuscript needs a some work in terms of its title and the addition of descriptive text focused on the internal process. If you have 3 characters, then there should be a corresponding amount of text describing the third character as you have devoted to the first two. I do not think you need additional figures, analyses, etc. You merely need to add text to the manuscript to give 1/3 of your character analysis the credit it deserves. And finally, I would advise a title change to reflect the scale of your dataset - it is not all lizards. it is some lizards, and a small sample at that. There is nothing wrong with that limited dataset because we have collected is important and of value and deserves to be published in PEERJ. The manuscript needs to be more specific about the limited data set and point out what is missing. And finally, you use the words Squamate, Lizard and Lacertilia, as well as Serpentes, throughout the manuscript, and in a variable manner - I hasten to remind you that snakes are lizards and your own phylogenetic diagrams make that clear. In any sense of lizard and squamate, they are thus the same thing, and in all cases, snakes are all three...squamates, lizards and lacertilians. This is merely a pet peeve of mine and reflects a general problem in herpetology, not one that is specific to your collaboration nor this manuscript.

I am opposed to anonymity in the peer review process and identify myself as Michael Caldwell.

·

Basic reporting

The authors present the morphological description of the middle ear in lizards based on clear stained specimen (of 38 lizard species) and discuss a possible ancestral state condition for the lizard middle ear plotting the described characters on a phylogenetic analysis based on molecular data. The manuscript is overall well prepared and well-written. In few occasions the structure of the sentences need tweaking. The methods and results sections are well organized and the discussion is straight forward. 5 tables and 8 figures are provided to present the data and illustrate the findings.

As shown by the authors (literature used is sufficient) data on the morphological shape of the different elements of the lizard middle ear is limited and a majority of the descriptions that are available are from older studies (60ies and 70ies). A fresh view and new descriptions on especially the extracolumella with new findings is of general interest to the scientific community and should be considered to be published.

However, there are some things in the manuscript that need attention: 1) The abstract should include the major findings of the study, in its current state it very general (see my comments in the attached pdf), 2) all figure captions need more explanations (what does the reader see, which view is presented, etc.), they are all in all very short and should be written is full sentences, 3) only figure 1 has a scale, but it is not indicated if that scale is the same for all illustrations seen in figure 1, also the other figures need a scale, 4) some microscopic photos are very dark and somewhat blurry (Figure 4 A and C, 5 C).

Experimental design

The basic experimental frame of the study is overall ok, one thing I see that seems problematic is that in most cases only one specimen of a species (23 of 38) and sometimes two and more were used for the clear staining part of the study. I understand that it is a quite long process to clear stain and needs time, but with only one specimen it is hard to tell anything about the variation within a species (as mentioned by the authors themselves in the manuscript). It is clear that this is not the point of the study, but it is a weak point when for example a general pattern for the extracolumella of a species is established.

I am mostly ok with the analytical part (software and approach) of the ancestral state reconstruction. But I suggest to revise the way character states were chosen in terms of the extracolumella. The authors erected four character states of the extracolumella for the ancestral state reconstruction and describe the morphology in great detail, because this is the structure with the most variance in the lizard middle ear. However, the character states that were chosen can lead to difficulties in following the arguments in the discussion. And here is why:
- The four states chosen by the authors are “reduced” [character state 0], “expanded” [character state 1], “absent” [character state 2], and “extensive” [character state 4].
- 1) the order seems odd. Although it probably will not influence the conclusions, I suggest to rethink the order and wording. Why would the state “absent” be the third state and “reduced” be the first? It does not follow a logic frame. There is the basic rule to look for characters to be present (and if present how they look) or absent. So, in this manner “absent” should be last in line.
- 2) the difference between the states “expanded” and “extensive” is not really clear. Does it differentiate between numbers of processes, or does it describe the size of the processes? This needs clarification.
- 3) the word “reduced” is difficult in relation with a character state, because it already implies some kind of modification of a previously more complicated structure and therefore cannot be the ancestral state. To avoid this the authors could just say “simple”.
- 4) the terms used for the states if the structure is present should follow a logical order from simple to complicated (especially helping readers who are not familiar with the type of analysis). So, I suggest to rethink the character states and maybe use different terms. For example: simple [0], expanded [1], extensive [2], absent [3]

The last point I want to address is the discussion, which is well written and detailed in its presentation. But I would like to see some kind of comparison of the findings of the ancestral state that was reconstructed in this analysis with what is available from the fossil record. I am aware there is not much, but it is worth looking in to the chapter “The lepidosaurian ear: variations on a theme” by Susan Evans (Springer Handbook of Auditory Research, Volume 59) for a summary of the fossil evidence. It’s always good to back up a theoretical ancestral state with actual evidence.

Validity of the findings

Although the middle ear of lizards is a quite well investigated subject the presented study adds new information. The authors make clear what the new findings are and discuss them with the available data from previous work. However, their important new insights are not summarized in the abstract in its current state.

The data are made available in 5 tables (4 tables intended for supplement) and 8 figures. The analysis appears sound but as mentioned earlier, the defintion of the character states needs attention.

The conclusion section needs revision as its lacks the major findings of the study in terms of the ancestral state reconstruction (see my comments in the attached pdf). However, the discussion is well supported with literature and research questions addressed in the introduction were matched.

---

## Round 0.2 · Minor Revisions

· Academic Editor

Minor Revisions

Dear authors,

Thank you for your revisions. Both reviewers feel that the paper is publishable pending corrections to a number of English grammar issues that are pervasive throughout the manuscript. Unfortunately, PeerJ does not provide editing as a standard service. However, may I suggest that you contact a fluent English-speaking colleague to look through the draft and make the necessary grammatical changes. Otherwise, there are a number of external editing services that are available.

Once your paper has been corrected along these lines, I will read through it to ensure that it is in acceptable English, but it will not require additional review by reviewers.

Thank you for your submission to PeerJ. I look forward to seeing it published. Please contact me if you have any questions.

Best,

Brandon P. Hedrick, Ph.D.

·

Basic reporting

The language of the paper still needs revision for English grammar, syntax, etc. You will see in my marked copy that I tackled the abstract but realized swiftly that my role as reviewer was not to perform this editing function for the entire manuscript.

The references, background, structure, figures, etc., are all fine.

Experimental design

I have read through the authors rebuttals to my earlier critiques and find that they addressed or rebutted everything to my satisfaction. The manuscript is retitled to reflect the experimental design, as are the explication of results and the discussion and conclusions.

As a second read, I find the manuscript in order on these points.

Validity of the findings

Same as above.

Additional comments

I would implore the authors to reread their manuscript in detail for English grammar, syntax, vocabulary, etc.

Other than that, the manuscript is scientifically sound.

·

Basic reporting

This is the revised version of a previously submitted manuscript. The manuscript was revised and reworked and is in great condition. Methods and results sections are well organized and the discussion was extended significantly and the abstract was reworked completely. The authors made a great decision in revising their character states and added more information on the extracolumella. The figures were reworked according to the reviewer’s suggestions.
The manuscript now is in great condition and suitable for publication. I am happy with all changes that were made.

Experimental design

Experimental design was good from the beginning, but the authors made adjustments to the definition of their character states as suggested by the reviewer. Ancestral state reconstruction ok. The revised version is in great condition.

Validity of the findings

Expanding the discussion and reworking the abstract significantly helped the previous version of the manuscript. I am happy with all changes and have nothing to add.

---

## Round 0.3 · Minor Revisions

· Academic Editor

Minor Revisions

Dear authors,

Thank you for your changes. I have just read through the manuscript and edited it again for English grammar issues and have found the following issues below that appear to have been missed by your colleague. Once you correct these, I'll move the paper to accept. There are just a few too many changes to take care of in the proof process.

Thanks for your submission. It is quite interesting and your figures are wonderful.

Best,

Brandon P. Hedrick, Ph.D.




Line 65: ‘although the number of taxa sampled in this study..’

Line 115: ‘the structures from which they originate’

Line 144: This sentence doesn’t make sense and I’m not sure how to fix it. Do you mean ‘unique middle ear morphologies present in diapsid and synapsid taxa’?

Line 159: What do you mean by main taxonomic groups? Can you add them in parentheses here to clarify?

Line 187: probably should be ‘s.p.’?

Line 411–12: ‘caused by inadequate specimen preparation’

Line 423: ‘extracolumellar processes’

Line 550: ‘with the latter structure displaying…’

Line 619: ‘parsimony-based ancestral state…’

Line 660: ‘ligamentous fibers that split into’

Line 691: ‘Tretioscincus do not have an anterior process’

Line 759: ‘it is not possible’

Line 766: ‘In C. collaris, it seems like the…’? Also the rest of this sentence needs work, but I’m not sure what you’re trying to say.

Line 772: ‘that the internal process has an auditive’

Line 840: ‘To understand the..’

Line 841: What should be evaluated? Incomplete sentence.

Line 845: ‘of a simple extracolumella’

Line 893: ‘might be through the quadrate’

Line 932: ‘evolutionary history’

Line 941: Grammar issues in this sentence. I’m not sure what you’re trying to say.

Line 954: ‘that lie adjacent’

Line 955: ‘shows the derived condition’

Line 955: Improved is the wrong word. Perhaps ‘evolved’

Line 957: ‘morphologies’

Line 995: ‘show a large amount of variation than previously’

Line 997: delete ‘A’. Change to: ‘Significant morphological variation of the’

Line 1017: ‘processes’

---

## Round 0.4 · accepted · Accept

· Academic Editor

Accept

Dear authors,

Thanks so much for your quick corrections. I feel that this paper is now ready to move to the proof stage. There are a few minor grammar issues (below) that should be fixed during the proof stage prior to publication. Congratulations and thank you for your submission! I look forward to seeing this work published!

Best,

Brandon P. Hedrick, Ph.D.


Line 550: ‘latter’ rather than later

Line 692: remove ‘is absent’

Line 766: There are still issues with this sentence. Perhaps: ‘In C. collaris, it appears that the role of the internal process is related to protection rather than hearing’

Line 942: ‘differs from it’

Line 966: ‘larger’